# Monomethyl auristatin antibody and peptide drug conjugates for trimodal cancer chemo-radio-immunotherapy

Dina V. Hingorani[1,7], Michael M. Allevato[2,7], Maria F. Camargo[1], Jacqueline Lesperance[1], Maryam A. Quraishi[1], Joseph Aguilera[1], Ida Franiak-Pietryga[1], Daniel J. Scanderbeg[1], Zhiyong Wang[2], Alfredo A. Molinolo[3,4], Diego Alvarado[5], Andrew B. Sharabi[1,4], Jack D. Bui[3,4], Ezra E. W. Cohen[4,6], Stephen R. Adams[2], J. Silvio Gutkind[2,4] & Sunil J. Advani [1,4 ✉]

Locally advanced cancers remain therapeutically challenging to eradicate. The most successful treatments continue to combine decades old non-targeted chemotherapies with radiotherapy that unfortunately increase normal tissue damage in the irradiated field and have systemic toxicities precluding further treatment intensification. Therefore, alternative molecularly guided systemic therapies are needed to improve patient outcomes when applied with radiotherapy. In this work, we report a trimodal precision cytotoxic chemo-radio-immunotherapy paradigm using spatially targeted auristatin warheads. Tumor-directed antibodies and peptides conjugated to radiosensitizing monomethyl auristatin E (MMAE) specifically produce CD8 T cell dependent durable tumor control of irradiated tumors and immunologic memory. In combination with ionizing radiation, MMAE sculpts the tumor immune infiltrate to potentiate immune checkpoint inhibition. Here, we report therapeutic synergies of targeted cytotoxic auristatin radiosensitization to stimulate anti-tumor immune responses providing a rationale for clinical translational of auristatin antibody drug conjugates with radio-immunotherapy combinations to improve tumor control.

[1] Department of Radiation Medicine and Applied Sciences, University of California San Diego, La Jolla, CA 92093, USA. [2] Department of Pharmacology, University of California San Diego, La Jolla, CA 92093, USA. [3] Department of Pathology, University of California San Diego, La Jolla, CA 92093, USA. [4] UC San Diego, Moores Cancer Center, La Jolla, CA 92093, USA. [5] Celldex Therapeutics, Hampton, NJ 08827, USA. [6] Department of Medicine, Division of Hematology and Oncology, University of California San Diego, La Jolla, CA 92093, USA. [7] These authors contributed equally: Dina V. Hingorani, Michael M. Allevato. ✉email: sjadvani@ucsd.edu

Non-metastatic, locally advanced, cancers continue to drive patient mortality. For such patients, randomized clinical trials from the 1970s to 1990s unequivocally demonstrated the superiority of concurrently given chemotherapy with radiotherapy[1–7]. Rationales for combining focal ionizing radiation (IR) with systemic drugs to attack macroscopic disease include: (1) independent mechanisms of tumor kill by chemotherapy and radiotherapy decrease emergence of treatment resistant tumor clonogens and (2) certain drugs radiosensitize, i.e., increase IR-induced DNA damage and cell death[8,9]. While radiotherapy delivery techniques have become increasingly sophisticated to conform IR dose deposition to tumors, systemic drugs given with complex intensity-modulated radiotherapy (IMRT) remain non-targeted cytotoxins (i.e., cisplatin, taxanes, 5-FU)[10–12]. Although such concurrent chemotherapies improve irradiated tumor control and patient survival, these non-targeted cytotoxins inherently cause normal tissue collateral damage in the irradiated field and have systemic toxicities diminishing patient quality of life. Moreover, therapy-induced toxicities cause treatment delays and chemo-radiotherapy dose reduction negatively impacting tumor control and patient survival.

Antibody drug conjugates (ADCs) provide an approach to deliver more potent cytotoxins specifically to tumors while sparing normal tissues[13,14]. ADCs split the roles of tumor targeting and cell killing into two distinct molecular tasks. Targeting is achieved by antibodies recognizing cell surface receptors preferentially expressed on tumor cells. Tumor cell kill is mediated by the conjugated drug payload, i.e., warhead. Following antibody-directed binding to specified cell surface receptors, the bound warhead is selectively taken up within target-rich tumors followed by drug release. For biomarker selected cancer patients, clinical safety and efficacy is being established with a growing number of ADCs constructed with cytotoxic warheads that include anti-tubulins (auristatins, maytansinoids), calicheamicins (ozogamicin), and topoisomerase inhibitors (exetecans)[15–22]. Monomethyl auristatin (MMAE) is the most common ADC warhead conjugated to an expanding array of antibodies in clinical development for oncologic applications. In addition to its exquisite cytotoxicity, MMAE radiosensitizes by increasing IR-mediated DNA double-strand breaks in irradiated cells[23].

Within the last decade, immunotherapy has revolutionized oncology by achieving durable tumor control in subsets of patients with advanced cancers[24,25]. There is tremendous excitement in evaluating chemotherapies and radiotherapy as catalysts to stimulate anti-tumor immune responses and augment the efficacy of immunotherapies to wider patient populations[26–30]. The ability of radiotherapy to be spatially and temporally targeted and induce tumor kill has spawned much interest as a method to engage anti-tumor immune responses[29,31–33]. For example, IR has been shown to enhance immune checkpoint inhibitor cytotoxic T lymphocyte responses in pre-clinical models and is under evaluation in clinical trials[27,34,35]. Mechanistically, IR induces immunogenic cell death and potentiates immunotherapies.

Here, we show spatially targeted auristatin tumor radiosensitization increases tumor control, activates anti-tumor immune response and potentiates immune checkpoint inhibition. To rigorously demonstrate this, we attach radiosensitizing MMAE warheads to two different classes of tumor-targeted drug carriers (i.e., antibody and activatable cell penetrating peptides (ACPPs)) using identical drug conjugation linker chemistry allowing simultaneous evaluation of MMAE with fractionated radiotherapy in isogeneic resistant xenografts, syngeneic tumors, and an autochthonous tumor model. Importantly, tumor-directed MMAE in combination with IR achieves durable tumor control across murine models irrespective of MMAE coupling to antibody or cell penetrating peptide delivery vehicles. Using RAG1

knockout mice and CD8 T cell-depleting antibodies, we establish a role for adaptive immunity in mediating sustained irradiated tumor control with MMAE. While MMAE with IR are cytotoxic and result in initial tumor growth delay, CD8 T cells are necessary for durable tumor control of MMAE radiosensitization. Finally, MMAE and IR stimulation of tumor immune cell infiltration potentiates anti-PD-1 immune checkpoint inhibition. As a functional consequence, MMAE-radiosensitized tumor kill induces immunologic memory. Based on these results, we propose integrating auristatin ADCs together with radiotherapy and immunotherapy to spatially restrict the immediate cytotoxic tumor kill of chemo-radiotherapy and then selectively engage adaptive immune responses.

## Results

**Spatially targeting MMAE with tumor-directed antibody and peptides.** We first evaluated the cytotoxicity of MMAE in a panel of human and murine cancer cellular systems. Compared to conventional chemotherapies cisplatin and paclitaxel, MMAE had increased potency across all human and murine cancer cell lines tested (Fig. 1a). While MMAE was cytotoxic to both human and murine cancer cells, we consistently observed that MMAE had increased potency in human cells compared to mouse lines. MMAE is a synthetic derivative of dolastatin 10 that inhibits tubulin polymerization[36]. As expected, blocking such an integral cellular process produced altered gene expression across major functional pathways in MMAE-treated cells (Fig. 1b, Supplementary Fig. 1a, b, Supplementary Data 1). By gene set analysis (GSA), the majority of pathways (17/25) were downregulated by MMAE, with the lowest directed enrichment scores involving autophagy, hypoxia, and metabolic stress pathways. Curiously, pathways with a positive directed enrichment score in tumor cells treated with MMAE involved extracellular tumor remodeling and included angiogenic signaling, matrix remodeling/metastasis and interferon signaling. To restrict MMAE toxicity to tumors and their associated immune microenvironment, we utilized parallel strategies to target drug delivery to tumors by simultaneously conjugating MMAE to tumor homing antibodies and cell penetrating peptides[23,37]. First, MMAE was coupled to the maleimidocaproyl-valine-citrulline-*p*-aminobenzyl carbamyl (MC-VC-PABC) linker[38]. The MC-VC-PABC linker is used in a clinically approved ADC (Fig. 1c)[15]. Another key rationale for use of MC-VC-PABC linkers for drug conjugation is that upon receptor-mediated internalization, the intervening valine-citrulline dipeptide is cleaved by endolysosomal cathepsin B followed by self-immolative loss of *p*-iminoquinone methide and $CO_2$ to release free drug which can be quantitated in tissues by LC–MS/MS analysis[39].

We synthesized a HER3-targeted auristatin ADC by reacting MC-VC-PABC-MMAE to anti-HER3 antibody CDX3379[40,41]. We choose HER3 as a cell surface target for ADCs for two reasons. First, an advantage of CDX3379 is that it recognizes both human and murine HER3 receptors allowing the study of MMAE warheads simultaneously in both human tumor xenografts and murine syngeneic tumor systems[42]. Second, HER3 has proven to be a bona fide cell surface ADC receptor target in clinical testing[43]. Based on prior studies optimizing drug:antibody ratio, 4 molecules of MMAE were attached per antibody and then the ADC labeled with Cy5 to allow for in-vivo tracking (Fig. 1d, Supplementary Fig. 2a, b)[44]. To test CDX3379-MMAE HER3 targeting in murine models, we used a panel of human and murine cancer cells expressing HER3 (Fig. 1e, Supplementary Fig. 1c). In immune-deficient mice with HER3 expressing human tumor xenografts, intravenously (i.v.) injected CDX3379-MMAE localized to tumor bearing hindlimbs (Fig. 1f, Supplementary Fig. 3a, b). Microscopic examination of tumor and adjacent normal muscle demonstrated

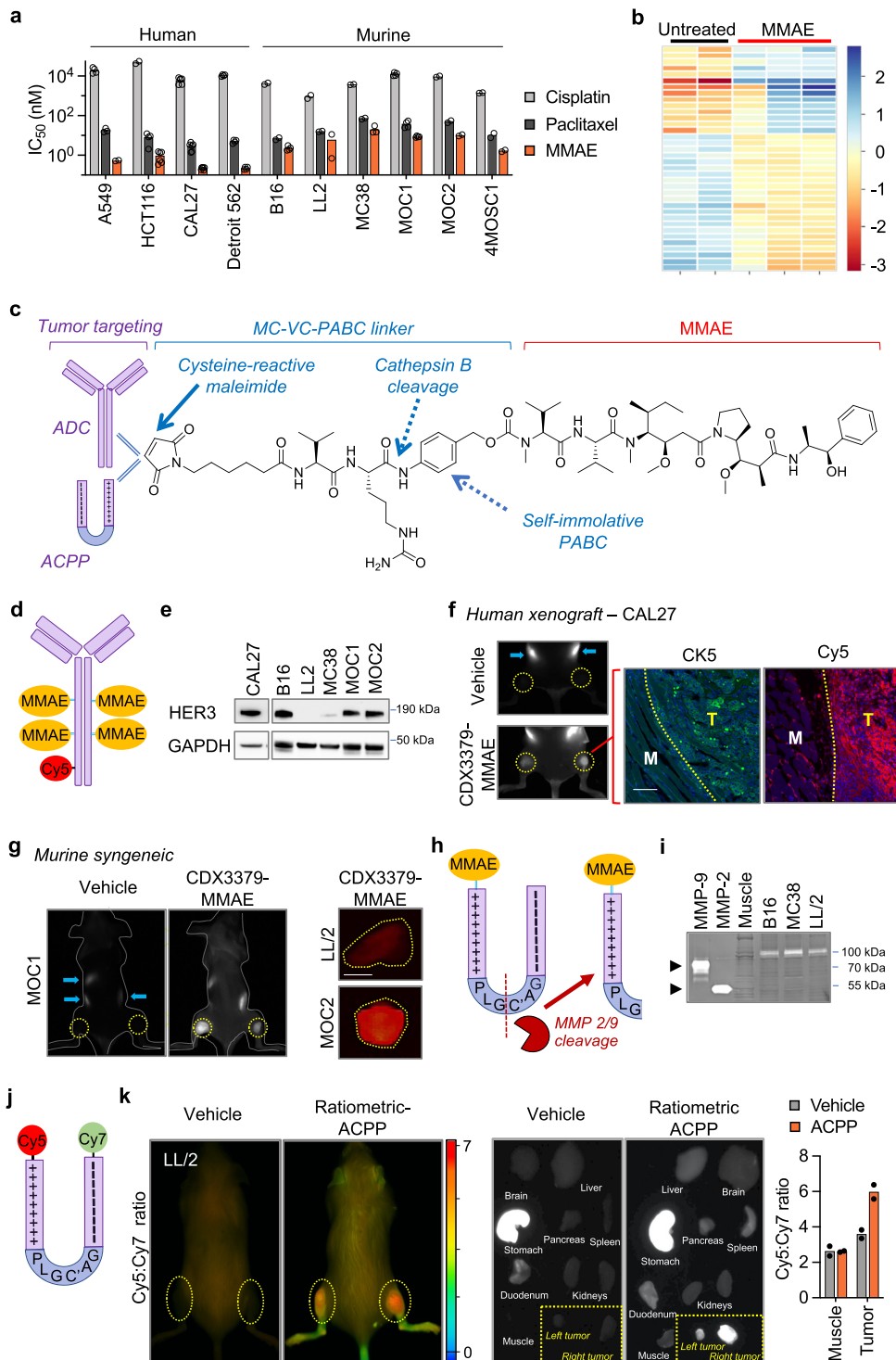

Cy5 labeled CDX3379-MMAE preferentially localized to tumor xenografts (CK5+ staining) while avoiding adjacent normal tissue (Fig. 1f). Importantly, i.v. injected CDX3379-MMAE also localized to HER3 expressing murine syngeneic tumors grown subcutaneously in the hindlimb (MOC1 and MOC2 tumors) or orthotopically in the tongue (MOC2 tumors) of immune-competent mice but not in tumors grown from HER3 negative LL/2 cells (Fig. 1e, g, Supplementary Fig. 3c, d).

There are limited pre-clinical studies evaluating efficacy of MMAE within the immune context in part due to ADC optimization of antibodies binding human receptors and paucity of receptor driven syngeneic cancer models[45]. As a solution to

test MMAE radiosensitization in commonly used pre-clinical syngeneic cancer models, we employed activatable cell penetrating peptide (ACPP) technology as a parallel approach for evaluating MMAE warheads in syngeneic tumors grown in immune-competent mice. ACPPs are designed with modular architecture and minimally consists of 3 domains: (1) a polycationic cell penetrating peptide (nine repeats of d-arginine, r9); (2) a polyanionic autoinhibitory domain (nine repeats of D-glutamic acid, e9); and (3) an intervening protease-sensitive flexible peptide linker (Fig. 1h, Supplementary Fig. 4)[23,46]. Importantly, drugs are coupled to the polycationic end of ACPP using the exact same warhead-linker chemistry as ADCs giving us

**Fig. 1 Spatially targeting MMAE to syngeneic murine tumors. a** $IC_{50}$ of cisplatin, paclitaxel and MMAE in human and murine cancer cell lines by Alamar Blue assay. Scatter plot with mean of biological replicates, number of biological replicates are provided in the Source Data file. **b** Heatmap depicting differential gene expression of control ($n = 2$) and MMAE ($n = 3$)-treated B16 cells. RNA expression analyzed using NanoString PanCancer IO 360 Panel. **c** Schema of antibody and cell penetrating peptide conjugates of MMAE using MC-VC-PABC linkers. **d** Structural representation of Cy5-labeled ADC with 4 MMAE molecules and 1 Cy5 per antibody. **e** HER3 expression in cancer cell lines. Immunoblot for total HER3 and GAPDH. **f** Cy5 localization of CDX3379-MMAE in subcutaneous tumor xenografts (yellow circles indicate tumor locations). 2.5 nmoles of Cy5-labeled CDX3379-MMAE or saline vehicle injected and whole mouse imaged for Cy5 (white signal) at 48 h. Gut auto-fluorescence indicated by blue arrows. Tissue sections containing tumor and adjacent normal muscle stained for CK5 (green) and imaged for Cy5 (red). Tumor (T) and normal muscle (M) boundary (yellow dotted line) delineated by CK5, scale bar = 100 μm. **g** Spatial localization of CDX3379-MMAE in subcutaneous syngeneic tumors. 2.5 nmoles of Cy5-labeled ADC or saline vehicle injected into tumor bearing mice and imaged at 48 h. For MOC1 tumors (yellow circles indicate tumor locations), whole mouse imaged for Cy5 (white signal). Imaged whole mouse delineated by dashed gray line. For LL/2 and MOC2 tumors, tumors resected (yellow dashed lines) and imaged for Cy5 (orange signal), scale bar = 5 mm. **h** Structural representation of ACPP-MMAE drug conjugate. MMP 2/9 cleavage site indicated by dashed red line. **i** Gelatin zymography of syngeneic murine tumor and adjacent muscle tissues. MMP-2 and MMP-9 standard activity in far-left lanes. **j** Structural representation of ratiometric ACPP probe with Cy5 and Cy7 attached to the polypeptide ends of ACPP. **k** Mice with subcutaneous LL/2 tumors (yellow circles indicate tumor locations) injected with 10 nmoles ratiometric ACPP or water vehicle. In situ whole mouse Cy5 and Cy7 imaging with Cy5:Cy7 emission ratio calculated (pseudocolor scale bar shown). Harvested tissues Cy5 imaging (white signal) shown and mean Cy5:Cy7 ratio scatter plot, $n = 2$ tissues. Images in (**e**, **i**) representative of three independent experiments. Source data provided in Source Data file.

the ability to interchangeably attach amine containing drugs such as MMAE to either ACPPs or ADCs through MC-VC-PABC linkers (Fig. 1c, Supplementary Fig. 5). Intact, ACPP scaffolding cloaks the conjugated drug in a pro-drug inaccessible state due to charge neutralization by the polyanionic peptide end which blocks intracellular uptake of the drug-conjugated polycationic cell penetrating peptide. Cleavage of the intervening ACPP linker activates the cell-penetrating peptide by removing the spatial proximity of the inhibitory polyanionic charge which allows for intracellular uptake of the drug-conjugated cell-penetrating peptide. Since tumors are preferentially enriched in extracellular matrix metalloproteinase (MMP) activity (Fig. 1i)[47], ACPP tumor targeting is achieved by inserting a MMP-2/9 sensitive PLGC(Me) AG peptide between the two charged domains of ACPP (Fig. 1h)[48]. Using a ratiometric ACPP probe co-labeled with Cy5 and Cy7 (Fig. 1j, Supplementary Fig. 6a), whole animal imaging established that i.v. injected ACPP is selectively cleaved and accumulates within routinely used pre-clinical LL/2 and MC38 murine syngeneic cancer models in situ (Fig. 1k, Supplementary Fig. 3e)[49,50]. Individual examination of fluorescent signaling in harvested tissues demonstrated ACPP preferentially localized to tumors over surrounding normal muscle tissue or other organs. In water vehicle injected mice, the stomachs had elevated Cy5 signal control mice due to gut/diet auto-fluorescence.

**Efficacy of MMAE in murine cancer models.** For in-vivo therapeutic studies, we initially characterized blood clearance of CDX3379-MMAE in immune-deficient and immune-competent mice using the attached Cy5 label as surrogate marker. After an initial Cy5 signal spike, blood Cy5 signal exponentially decreased and returned to background fluorescence by day 7 post i.v. injected Cy5-labeled CDX3379-MMAE compared to blood from saline injected control mice (Fig. 2a). We then validated CDX3379-MMAE released its conjugated drug warhead by quantitating free MMAE accumulation in tumor tissues by LC–MS/MS compared to control tumors from saline-injected mice (Fig. 2b). Since MMAE is a potent tumoricidal drug and studied primarily for ADC monotherapy, we evaluated the intrinsic tumoricidal activity of CDX3379-MMAE. First, we tested direct MMAE tumor kill by using isogeneic cell lines with differential sensitivity to anti-HER3 antibody due to expression of a PIK3CA point mutant[42]. Orthotopic tongue tumor xenografts were established from CAL27 wild-type (WT) or PIK3CA H1047R-expressing cells. In parental WT CAL27 tumors, a single 2.5 nmole injection of CDX3379 antibody produced significant growth delay by as assessed by both gross macroscopic tumor

volume response and histologic Ki-67 staining, that was further enhanced by CDX3379-MMAE conjugate (Fig. 2c, Supplementary Fig. 7a). As expected, PIK3CA H1047R-expressing tumors were resistant to CDX3379 antibody which was not overcome by increased antibody dose delivery (Fig. 2c, Supplementary Fig. 7b). In contrast, CDX3379-MMAE conjugate produced significant growth delay in these HER3-resistant tumors. Taken together, these results validate the advantage the MMAE warheads to bypass receptor signaling resistant mechanisms and kill cells irrespective of sensitivity to antibody receptor signal inhibition.

We then advanced testing MMAE in immune-competent models. Mice with syngeneic MOC2 cells orthotopically grown in tongues were given a single 2.5 nmole i.v. injection of CDX3379 antibody or CDX3379-MMAE ADC. CDX3379 antibody lacked efficacy while CDX3379-MMAE produced modest tumor growth delay (Fig. 2d). Tumors were harvested and immune infiltrate profiled by CyTOF analysis[51] (Fig. 2d, Supplementary Fig. 8). CDX3379 antibody significantly increased the M2 macrophage population that was also seen with CDX3379-MMAE indicating an antibody-driven effect. Specific immune populations that increased with the conjugated MMAE warhead included regulatory T cells, dendritic cells and NK cells that were upregulated following CDX3379-MMAE injection but not by CDX3379 antibody alone. To rigorously evaluate CDX3379-MMAE in HER3-expressing syngeneic models, we repeated studies in subcutaneously grown MOC1 and B16 tumors (Fig. 1e, Supplementary Fig. 3d). Similar to our findings in orthotopic MOC2 tumors, CDX3379-MMAE showed modest anti-tumor activity as measured by mouse survival in both MOC1 and B16 tumors (Fig. 2e). To establish CDX3379-MMAE specificity, we synthesized a cetuximab-MMAE conjugate[37]. Cetuximab is a clinically approved anti-EGFR antibody to the human isoform. While Cy5-labeled CDX3379-MMAE showed dose response cell surface binding to B16 cells, cetux-MMAE did not appreciably attach to B16 cells (Fig. 2f). While CDX3379-MMAE showed efficacy, cetux-MMAE did not significantly increase survival in B16 tumor bearing mice when compared to untreated control mice (Fig. 2e). Finally to evaluate MMAE independent of antibody effects or delivery limitations, we coupled MMAE to the ACPP drug carrier scaffold to create ACPP-MMAE (Supplementary Fig. 5). Spatial tumor targeting of i.v. injected ACPP was validated with a Cy5-labeled probe that selectively localized to syngeneic 4MOSC1 tumors grown orthotopically in murine tongues (Fig. 2g, Supplementary Fig. 6b). Concordant to our results with antibody conjugation, ACPP-MMAE conjugate monotherapy produced modest tumor growth delay (Fig. 2h).

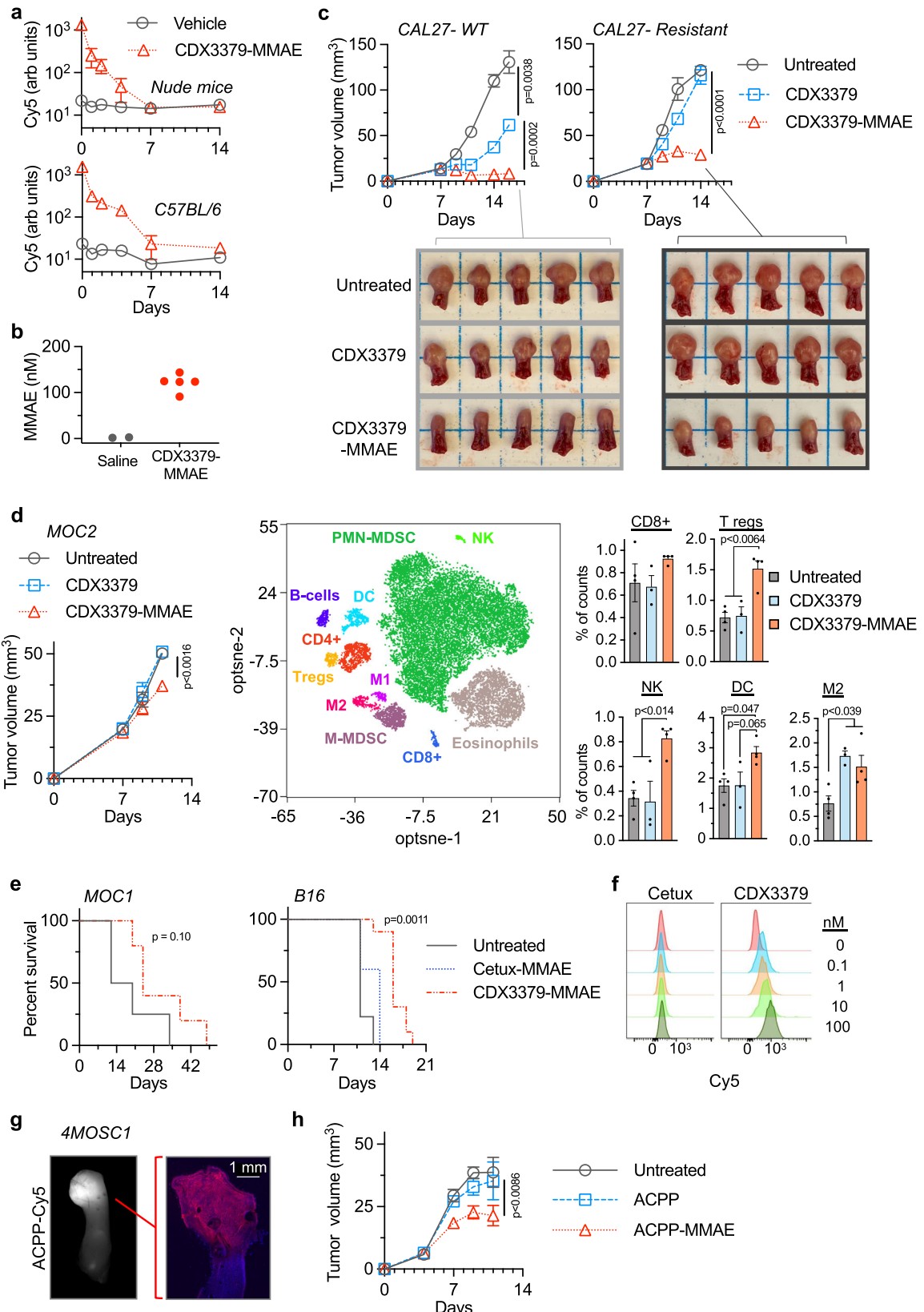

**MMAE radiosensitization produces durable tumor control**. MMAE is a potent cytotoxic anti-tubulin (Fig. 3a, b, Supplementary Fig. 9a) arresting cells in the radiosensitive mitotic phase of the cell cycle (Fig. 3c, Supplementary Fig. 9b)[23]. In addition to intrinsic cell kill, single nanomolar doses of MMAE-sensitized murine tumor cells to IR-mediated clonogenic cell death using clinically relevant 2 Gy doses (Fig. 3d, Supplementary Fig. 9c). Interestingly, MMAE down-regulated genes involved in DNA damage repair (Supplementary Fig. 1a). Given the modest anti-tumor activity of MMAE monotherapy in syngeneic murine tumors (Fig. 2), we evaluated MMAE within the irradiated tumor immune microenvironment. Since MMAE radiosensitizes and

**Fig. 2 Auristatin activity in murine models. a** Nude or C57BL/6 mice injected with saline or Cy5-labeled ADC on day 0. Blood Cy5 plotted as mean ± SD, $n = 3$. **b** Mice with orthotopic CAL27 tongue xenografts injected with 2.5 nmoles of CDX3379-MMAE ($n = 4$) or saline ($n = 2$) and tumors harvested 48 h. Tumor drug concentration scatter plot. **c** Mice implanted orthotopically with CAL27 parental (WT) or resistant (PIK3CA H1047R) cells on day 0 injected with 2.5 nmoles drug on day 7. Tumor volume plotted as mean ± SEM, $n = 6$. Statistical significances calculated using two-way ANOVA with Tukey's multiple comparisons test. Tumor burden in resected tongues. **d** Mice implanted with MOC2 cells on day 0 injected with 2.5 nmoles drug on day 7. Tumor volumes plotted as mean ± SEM, $n = 4$. Statistical significance calculated using two-way ANOVA with Tukey's multiple comparisons test. Tumors analyzed on day 11 by CyTOF using t-distributed stochastic neighboring embedding (tSNE). Concatenated image delineating immune cell discrimination. Scatter plot with mean ± SEM of untreated and CDX3379-MMAE, $n = 4$; CDX3379, $n = 3$. Statistical significances calculated using one-way ANOVA with Tukey's multiple comparisons test. **e** For MOC1 tumors, mice injected on day 0 (tumor volume ~40 mm³) with 2.5 nmoles CDX3379-MMAE ($n = 5$) or untreated ($n = 4$). For B16, mice implanted with tumor cells on day 0 and injected on day 3 with 5 nmoles CDX3379-MMAE ($n = 10$), cetux-MMAE ($n = 5$) or untreated ($n = 9$). Mouse survival plotted and statistical significance of CDX3379-MMAE to control or cetux-MMAE calculated using Log-rank (Mantel–Cox) test with single comparisons. **f** B16 cell surface binding of Cy5-labeled ADCs measured by flow cytometry and plotted as percent of maximum. **g** Cy5-labeled ACPP injected into mouse with orthotopic 4MOSC1 tumor, tongue resected, imaged for Cy5 (white signal, left image), sectioned (right image), nuclei DAPI stained (blue) and Cy5 captured (red). **h** Mice implanted with orthotopic 4MOSC1 cells on day 0 injected with 10 nmoles drug on days 5 and 7 ($n = 8$) or untreated ($n = 9$). Tumor volumes plotted as mean ± SEM. Statistical significance calculated using two-way ANOVA with Tukey's multiple comparisons test. Source data provided in Source Data file.

clinical radiotherapy is predominantly given in a fractionated manner over multiple days, we tested a regimen concurrently delivering two doses of MMAE with three fractions of IR (Fig. 3e). In pilot studies, LL/2 and B16 tumors exhibited relative resistance to IR compared to MC38 tumors. Therefore, we used a dose of 5 Gy per fraction for mice bearing LL/2 and B16 tumors and a lower 3 Gy dose per fraction for more sensitive MC38 tumors. Similar to our above findings with CDX3379-MMAE or ACPP-MMAE conjugate monotherapy in syngeneic murine tumor models (Fig. 2), MMAE monotherapy lacked significant anti-tumor efficacy in three well described syngeneic murine models (LL/2, B16, and MC38) when delivered to tumors using the ACPP scaffold[52] (Fig. 3e, Supplementary Fig. 9d). In contrast, MMAE combined with fractionated IR significantly increased tumor control across this diverse panel of syngeneic murine models albeit with varying degrees of efficacy. When comparing all four treatment groups by Log-rank (Mantel–Cox) test, the differences in survival were statistically significant, $p < 0.0001$. As radiosensitization effects are often normalized to drug alone effects, mice treated with ACPP-MMAE + IR had significantly improved survival when directly compared to ACPP-MMAE across all three tumor models, $p < 0.01$. In poorly immunogenic LL/2 tumors[52], ACPP-MMAE + IR modestly prolonged mouse survival and tumor control when compared to MMAE or IR monotherapies (Fig. 3e, Supplementary Fig. 9d). In contrast, combining MMAE and IR produced long-term survival and tumor control in B16 and MC38 tumor models that was not seen with MMAE or IR monotherapies. The variability in anti-tumor efficacy in cancer models was not due to differences in drug delivery since i.v. injected ACPP-MMAE efficiently released its conjugated drug warhead equally in LL/2 and B16 tumors (Fig. 3f). From a safety perspective, no adverse effects of ACPP-MMAE and IR on mouse well-being was observed by body weight (Supplementary Fig. 9d).

While syngeneic tumors grown from established cell lines in immune-competent mice provide immune context, a limitation of such models is that cells used to induce tumors undergo selection for growth in cell culture losing tumor heterogeneity and rapidly form tumors[53]. Moreover, tumor cell implantation induces tissue stresses. To more faithfully recapitulate the co-evolving tumor-immune interplay and remove tumor grafting issues, we tested MMAE and IR in an autochthonous carcinogen-induced model of oral cancer where mice are exposed to 4-nitroquinoline N-oxide (4NQO)-laced water (Fig. 3g)[54]. 4NQO chemical carcinogenesis mimics cigarette smoking induced DNA damage and the human tobacco-associated mutanome[55]. After prolonged exposure to 4NQO, mice reproducibly develop oral tumors including

on their tongues. C57BL/6 mice were exposed to 4NQO-laced water for 16 weeks and then maintained off 4NQO for 4 weeks. At week 20 after 4NQO initiation, mice were treated with i.v. injected ACPP-MMAE and focal tongue IR. At week 22, tongues were harvested and histologically assessed for total tumor burden by a pathologist in a blinded manner (Fig. 3g). Consistent with our results from established cell line models, combining MMAE with IR significantly decreased total tumor burden in this autochthonous model when compared to untreated control tumors.

Given the ability of ACPP delivered MMAE to radiosensitize syngeneic tumors, we then tested the ADC drug delivery platform. Using a regimen equivalent to ACPP-MMAE + IR dosing (Fig. 3e), mice with B16 tumors were treated CDX3379-MMAE and IR with the only difference being the dose of ADC reduced to 2.5 nmoles. This was done to maintain MMAE molar equivalence between the ADC and ACPP injections. On average, our synthesized CDX3379-MMAE have four MMAE molecules attached/antibody. In contrast, ACPP-MMAE has one MMAE attached/cell penetrating peptide carrier. Therefore to maintain equivalent MMAE molecules between ACPP and ADC drug-delivery platforms, we injected 2.5 nmoles CDX3379-MMAE to fairly compare it to 10 nmoles of ACPP-MMAE since both doses result in 10 moles of MMAE drug being i.v. injected. When combined with fractionated IR, CDX3379-MMAE also demonstrated prolonged tumor control and survival when compared to untreated control tumors, $p = 0.003$ (Fig. 3h). Importantly, CDX3379-MMAE + IR tumor control phenocopied ACPP-MMAE + IR in B16 tumors (Fig. 3e). Interestingly, there are 2 derivatives of auristatins that are in clinical development as ADC warheads, MMAE and monomethyl auristatin F (MMAF) (Fig. 3i). As free drug, MMAF has reduced cell permeability compared to MMAE resulting in less potency. However, an advantage of MMAF warheads is reduced off-target cell kill compared to antibody conjugated MMAE[39]. To compare auristatin derivatives radiosensitization in the irradiated tumor immune microenvironment, we attached MMAF to CDX3379 (CDX3379-MMAF) (Supplementary Fig. 2c, d). In combination with IR, CDX3379-MMAE proved to be superior to CDX3379-MMAF, $p = 0.04$ (Fig. 3h). Therefore, we focused further auristatin radiosensitization studies on the MMAE derivative.

**Adaptive immunity drives MMAE-radiosensitized durable tumor control.** To interrogate immune contributions of radiosensitized MMAE tumor control, we characterized how MMAE, and IR influenced tumor immune infiltration. First, tumor single

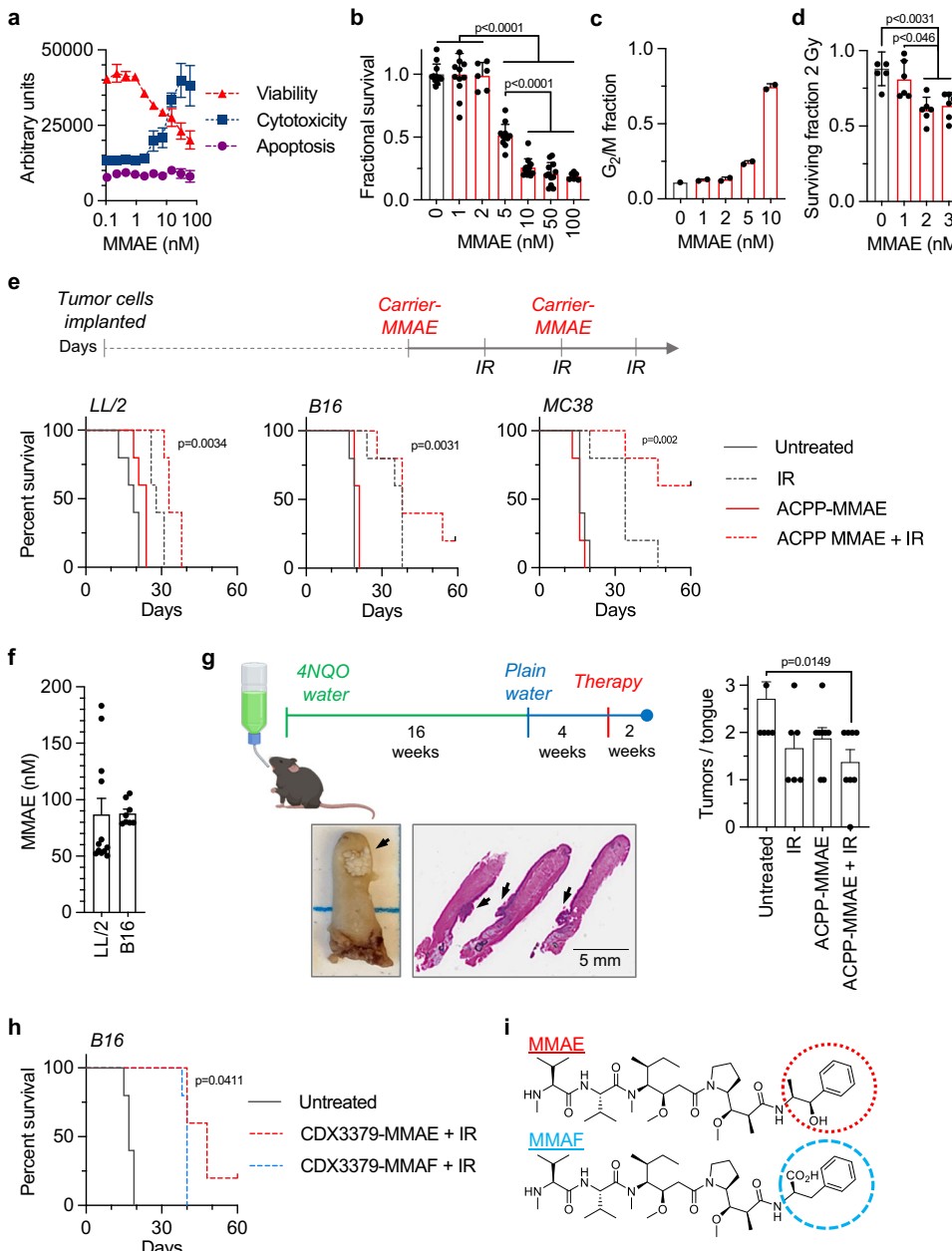

**Fig. 3 MMAE radiosensitizes syngeneic murine tumors. a** B16 cells treated with MMAE and analyzed by ApoTox-Glo Triplex Assay. Data plotted as mean ± SD, $n = 3$. **b** For clonogenic survival, B16 cells treated with MMAE, colony number normalized and plotted as mean ± SEM, $n = 11$ (0, 1, 5 10, 50 nM), $n = 5$ (2, 100 nM). Statistical significances calculated using one-way ANOVA with Tukey's multiple comparisons test. **c** Cell cycle analysis of MMAE-treated B16 cells. Data plotted as mean ± SD, $n = 2$. **d** Clonogenic survival of B16 cells treated with MMAE and 2 Gy. Data plotted as mean ± SEM, $n = 6$. Statistical significances calculated using one-way ANOVA with Sidak's multiple comparisons test. **e** Treatment scheduling MMAE + IR in subcutaneous tumors. Tumor cells implanted on day 0. Treatment started on day 3 for LL/2 and B16 and day 6 for MC38 tumors. 10 nmoles ACPP-MMAE injected for 2 doses. IR given on 3 consecutive days, 5 Gy/day (LL/2, B16) and 3 Gy/day (MC38). Mouse survival plotted and statistical significances (ACPP-MMAE + IR vs. ACPP-MMAE) calculated using Log-rank (Mantel–Cox) test, $n = 5$. **f** Tumor-bearing mice injected with 10 nmoles of ACPP-MMAE, $n = 12$ (LL/2) $n = 8$ (B16). Tumors harvested, drug concentration scatter plot with mean ± SEM. **g** 4NQO carcinogen-induced tongue model. At week 20 after 4NQO exposure, mice injected with 10 nmoles ACPP-MMAE × 2 doses. Whole tongue IR given as 5 Gy × 2. Tongues resected at week 22, H&E stained, and tumors quantified. Representative whole and sectioned untreated tongues (tumors indicated by black arrows). Mean tumors/tongue ± SEM plotted, $n = 7$ (untreated), $n = 6$ (IR), $n = 8$ (ACPP-MMAE, ACPP-MMAE + IR). Statistical significances calculated using one-way ANOVA with Tukey's multiple comparisons test. **h** Mice implanted with B16 tumor cells on day 0 and treated as in Fig. 3e except for 2.5 nmole dose of ADC. Mouse survival plotted and statistical significance (CDX3379-MMAE + IR vs. CDX3379-MMAF + IR) calculated using Log-rank (Mantel–Cox) test, $n = 5$. **i** Chemical structure of auristatins with moieties altering cell permeability circled. Results in (**a**, **c**) representative of 2 independent experiments. Source data provided in Source Data file.

cell suspensions were analyzed by flow cytometry (Fig. 4a, Supplementary Fig. 10a). As a percentage of live cells, CD45+ cells were 5.2%, 8.6%, 26.3%, and 69.1% for vehicle injected (water) control, ACPP-MMAE, IR and ACPP-MMAE + IR-treated

tumors respectively. Quantitating CD4+, CD8+, and NK cells within tumors demonstrated an increase of these immune cell populations when MMAE and IR were given together (Fig. 4a, Supplementary Fig. 10b). In particular, a striking relative and

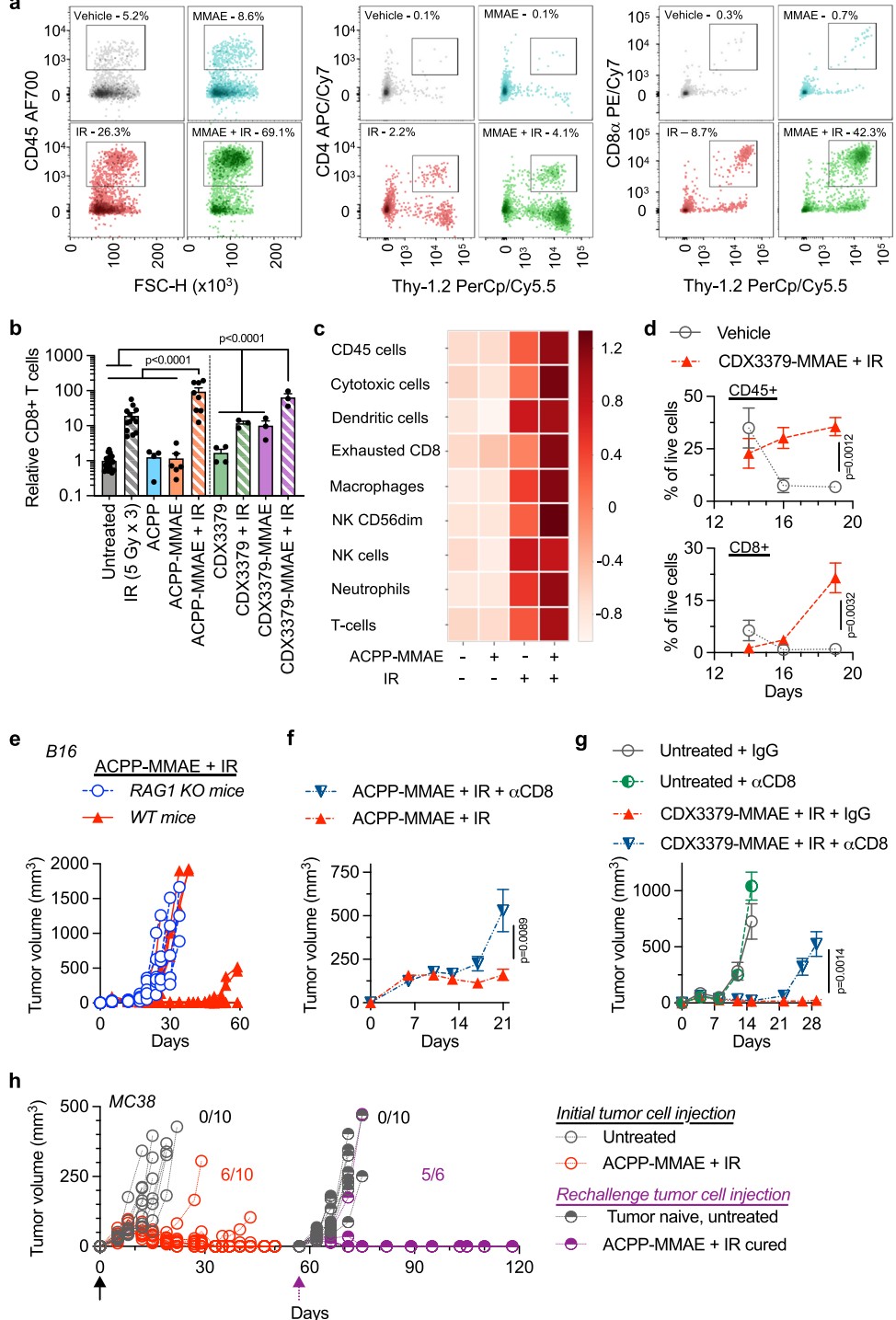

absolute increase in CD8 T cells was seen in MMAE-radiosensitized tumors. As a percentage of live cells, CD8+ T cells were 0.3%, 0.7%, 8.7%, and 42.3% for vehicle injected (water) control, ACPP-MMAE, IR, and ACPP-MMAE + IR-treated tumors, respectively. We confirmed that the control ACPP carrier (r9 cell-penetrating peptide without MMAE conjugated) had no influence on tumor CD8 T cell infiltration (Fig. 4b). To comprehensively investigate the tumor immune microenvironment in an unbiased manner, we employed Nano-String analysis (Fig. 4c, Supplementary Data 2)[55,56]. Compared to vehicle injected (water) control tumors, IR increased tumor infiltration of a variety of immune cell types while ACPP-MMAE

had minimal effects. Importantly, combining MMAE with IR further increased distinct immune cell populations within tumors, including cytotoxic cells. Since sustained tumor control with MMAE radiosensitization (Fig. 3e, h) was associated with increased CD8 T cell infiltration (Fig. 4a–c), we further characterized tumor CD8 T cells. Irradiation combined with MMAE increased CD8 T cell granzyme B at both the transcriptional and protein levels (Supplementary Fig. 10c). Given the activation of CD8 T cells in MMAE radiosensitized tumors with ACPP drug delivery, we tested if antibody-conjugated MMAE radiosensitization also stimulated CD8 T cell tumor infiltration. On day 9 post B16 tumor cell implantation, mice were treated as

**Fig. 4 MMAE radiosensitization stimulates immune responses. a** Mice with B16 tumors treated with ACPP-MMAE and IR per Fig. 3e starting on day 9 post implantation. Tumors harvested on day 19 and immune infiltrates characterized by flow cytometry. Results representative of three independent experiments. **b** B16 tumors treated as in Fig. 4a, CD8 tumor infiltration quantified by flow cytometry, normalized to untreated tumors, and plotted as mean ± SEM: untreated ($n = 18$), IR ($n = 13$), ACPP-MMAE + IR ($n = 8$), ACPP-MMAE ($n = 6$), ACPP, CDX3379 ($n = 4$), CDX3379, CDX3379-MMAE, CDX3379-MMAE + IR ($n = 3$). Statistical significances calculated using one-way ANOVA with Tukey's multiple comparisons test. **c** Mice with B16 tumors treated with ACPP-MMAE and IR per Fig. 3e starting on day 8. Tumors harvested on day 13 and analyzed using NanoString nCounter PanCancer Mouse Immune Profiling panel. Heatmap depicts Z-score of immune signatures, $n = 3$. **d** B16 tumors treated as in Fig. 3h starting day 9 post tumor implantation. Tumor infiltrates quantified by flow cytometry and plotted as mean ± SEM, $n = 3$ (vehicle day 14), $n = 4$ (all other points). Statistical significance calculated using two tailed t-testing with multiple comparisons. **e** B16 tumors implanted in WT or RAG1 KO mice on day 0 treated with ACPP-MMAE and IR as in Fig. 3e starting on day 3. Individual tumor volumes plotted, $n = 10$. Data from WT mice is from Fig. 3e. **f** Mice given anti-CD8 antibody (10 mg/kg). B16 tumor cells implanted on day 0 and given 10 nmoles ACPP-MMAE and 5 Gy as in Fig. 3e starting on day 3. Data plotted as mean ± SEM, $n = 10$. Statistical significance calculated using two-way ANOVA with Tukey's multiple comparisons test. **g** B16 tumor bearing mice given anti-CD8 antibody and treated with 2.5 nmoles CDX3379-MMAE and 5 Gy as in Fig. 4f. Data plotted as mean ± SEM and statistical significances calculated using two-way ANOVA with Tukey's multiple comparisons test, $n = 10$. **h** MC38 tumor cells implanted on day 0 and mice injected with 10 nmoles ACPP-MMAE and given 3 Gy as in Fig. 3e starting on day 6. On day 59, tumor cured and tumor naïve mice rechallenged with tumors. Individual tumor volumes plotted, $n = 10$ for all groups except ACPP-MMAE + IR cured, $n = 6$. Source data in Source Data file.

schemed in Fig. 3e with MMAE delivered as an ADC. The last day of IR was on day 11 and tumors were harvested on days 14, 16, and 19. Flow cytometry analysis showed a temporal increase in CD45+ cells and CD8 T cells in tumors from mice treated with CDX3379-MMAE + IR that was significantly elevated over vehicle injected (saline) control tumors at 10 days after the treatment package was completed (Fig. 4d). Importantly, CDX3379-MMAE + IR significantly increased CD8 tumor cell infiltration when compared to vehicle control, CDX3379 antibody alone, CDX3379-MMAE alone, IR alone or CDX3379 + IR-treated tumors (Fig. 4b) providing evidence that MMAE drug warhead and not the carrier antibody-stimulated immune infiltration of the irradiated tumor immune microenvironment.

Given the alterations of CD8 T cells both quantitatively and functionally, we interrogated the contribution of adaptive immune responses on anti-tumor efficacy of MMAE radiosensitization. First, we used a genetic approach where B16 tumors were grown in RAG1 knockout (KO) mice that lack mature B and T cells and compared them to immune-competent mice[57]. Tumor-bearing RAG1 KO mice were treated with ACPP-MMAE and IR in an identical regimen used in tumors grown in WT mice. ACPP-MMAE or IR monotherapy had similar tumor growth curves and mouse survival irrespective of whether tumors were grown in RAG1 KO or WT genetic backgrounds (Fig. 3e, Supplementary Fig. 10d). However and in direct contrast to the durable tumor control seen of MMAE-radiosensitized tumors in WT mice, ACPP-MMAE together with IR did not appreciably increase tumor control in RAG1 KO mice compared to IR alone. Specifically comparing tumor growth of mice treated with the combination of ACPP-MMAE and IR, a proportion of tumors from WT mice demonstrated durable tumor control while none of the tumors grown in RAG1 KO were controlled by day 34 (Fig. 4e). Since adaptive immune responses were required for prolonged tumor control of concurrently given MMAE and IR, we examined the necessity of CD8 T cells. WT mice were injected with anti-CD8 T cell-depleting antibody and then treated with ACPP-MMAE + IR (Fig. 3e). For the initial 2–3 weeks, mean tumor volume responses were similar irrespective of CD8 T cell status (Fig. 4f). However after day 21, CD8 T cell-depleting antibody abrogated durable tumor control of concurrent MMAE + IR. We then corroborated these findings with the ADC drug-delivery platform. Importantly, CDX3379-MMAE + IR phenocopied tumor volumetric response observed with ACPP-MMAE + IR in mice given a control IgG antibody (Fig. 4g). As with ACPP-MMAE and IR, CDX3379-MMAE + IR produced initial tumor control irrespective of CD8 depletion. However by day 28, tumors from CD8 T cell-depleted mice

treated with CDX3379-MMAE and IR were significantly larger than those from mice injected with a control IgG antibody.

Finally, we tested if tumor kill mediated through MMAE radiosensitization-induced immunologic memory. MC38 tumor-bearing mice were treated with ACPP-MMAE + IR as schemed in Fig. 3e. All untreated tumors grew out by day 22 and required mice to be sacrificed (Fig. 4h). In contrast, ACPP-MMAE + IR produced tumor control in the majority of mice. In three mice, the bilateral treated tumors were completely eradicated and on day 59 these three mice were re-challenged with bilateral MC38 tumor cell implantation. Of six tumors implanted, five tumor injection sites showed no evidence of tumor growth in the subsequent 60 days (day 120 from initial tumor cell implantation). As a control for tumor cell implantation on day 59, 5 tumor-naïve mice were also injected bilaterally with MC38 tumor cells on day 59 and all 10 tumors grew out. No adverse effects of ACPP-MMAE and IR on mouse well-being was observed by body weight (Supplementary Fig. 9e). Taken together, tumor directed MMAE + IR produced initial tumor cell kill and tumor growth delay which is associated with increased tumor CD8 T cell infiltration. While the initial cytotoxic effects of IR and MMAE delayed tumor growth, CD8 T cells were required to achieve durable tumor control of MMAE radiosensitized tumors and induced immunologic memory against tumor cells. Moreover, MMAE radiosensitized immunogenic tumor control occurred irrespective of how MMAE was targeted to tumors as we found similar results by using a cell-penetrating peptide (ACPP) or antibody (ADC) carrier to deliver MMAE to tumors suggesting broad applicability of MMAE-mediated tumor radiosensitization.

**Immune checkpoint inhibition potentiates MMAE radiosensitization.** Unbiased NanoString analysis revealed increased CD8 T cell exhaustion in irradiated tumors (Fig. 4c). Therefore, we further characterized the immune checkpoint axis and found PD-L1 and PD-1 were up-regulated in irradiated tumors ± MMAE (Fig. 5a). Given the modulation of T cell exhaustion and necessity of CD8 T cells for durable tumor control of MMAE radiosensitization, we tested if immune checkpoint inhibition potentiated tumor control in combination with MMAE and IR. For immune checkpoint inhibition, anti-PD-1 antibody therapy was initiated together with the first MMAE injection and a total of three injections of anti-PD-1 antibody were given[26]. Concordant with our above results (Fig. 3e), IR alone produced tumor growth delay that was augmented by the addition of MMAE in B16 tumors (Fig. 5b, top half). Interestingly, immune checkpoint inhibition with anti-PD-1 antibody did not have appreciable therapeutic effect when given with MMAE or 5 Gy fractionated

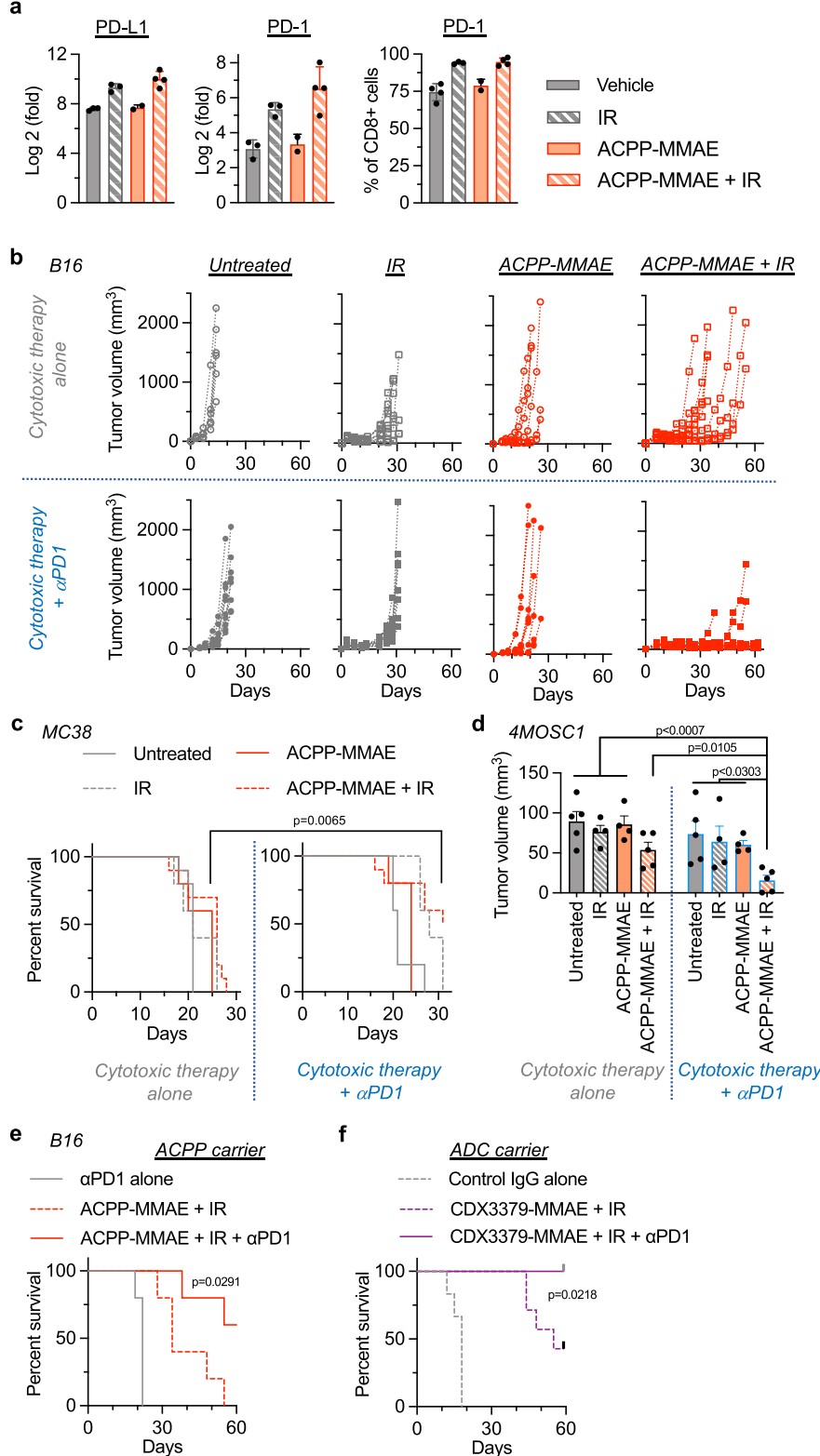

IR monotherapies (Fig. 5b, lower half). However, adding anti-PD-1 antibody specifically to the combination of MMAE + IR produced durable tumor control in the majority of tri-modality treated tumors (Fig. 5b).

Importantly, we validated these results in two additional syngeneic cancer models (MC38 and 4MOSC1) using the exact same temporal dosing schedule of anti-PD-1 therapy. Anti-PD-1 antibody was started simultaneously with the first injection of

MMAE and given in 3 total doses over a course of 8 days. Since MC38 tumors showed significant response to MMAE + IR concurrent therapy without immune checkpoint inhibition (Figs. 3e and 4h), we started with larger MC38 tumors in studies evaluating tri-modality therapy. Instead of initiating MMAE on day 6 post tumor cell implantation as in the above studies with MC38, tri-modality therapy was initiated on day 10 post tumor implantation which resulted in larger tumors that proved more

**Fig. 5 Checkpoint inhibition potentiates MMAE radiosensitization. a** Mice with B16 tumors treated with vehicle, IR ($n = 3$) ACPP-MMAE ($n = 2$), or ACPP-MMAE + IR ($n = 4$) as in Fig. 4a. Tumor RNA analyzed by qPCR for PD-L1 and PD-1. CD8 PD-1 expression analyzed by flow cytometry. Scatter plot with mean ± SEM. **b** Mice implanted with B16 tumor cells on day 0 and injected with 10 nmoles ACPP-MMAE on day 3 and 5 Gy as in Fig. 3e. Anti-PD-1 antibody (200 µg) given i.p. on days 3, 6, and 11 (lower row). Individual tumor volumes plotted, $n = 6$ (untreated, ACPP-MMAE alone), $n = 10$ (all other groups). **c** MC38 tumors implanted on day 0 and mice injected with 10 nmoles ACPP-MMAE and 3 Gy as in (**b**) starting on day 10. Anti-PD-1 antibody injected on days 10, 13, and 18 (right panel). Mouse survival plotted and statistical significances (ACPP-MMAE + IR vs. ACPP-MMAE + IR + anti-PD-1 antibody) calculated using Log-rank (Mantel–Cox) test, $n = 10$ (untreated, ACPP-MMAE + IR ± PD1), $n = 5$ (all other groups). **d** Mice with orthotopic oral 4MOSC1 tumors implanted on day 0 injected with 10 nmoles ACPP-MMAE on day 5 and 2 Gy on day 6. Anti-PD-1 antibody given on days 5, 8, and 13 (right panel). Tumor volume scatter plot with mean ± SEM. Statistical significances calculated using two-tailed $t$ tests without multiple comparisons, $n = 5$ (untreated, ACPP-MMAE + IR); $n = 4$ (IR, ACPP-MMAE) 4). **e** C57BL/6 mice with B16 tumors implanted on day 0 injected with 10 nmoles ACPP-MMAE and 5 Gy as in (**b**) starting on day 3. Anti-PD-1 antibody given on days 3, 6, and 11. Mouse survival plotted and statistical significances (ACPP-MMAE + IR vs. ACPP-MMAE + IR + anti-PD-1) calculated using Log-rank (Mantel–Cox) test, $n = 5$. Survival data from mice in (**b**). **f** C57BL/6 mice with B16 tumors implanted on day 0 treated with 2.5 nmoles CDX3379-MMAE, 5 Gy and anti-PD-1 antibody as in (**e**) starting on day 3. Mouse survival plotted, $n = 6$ (Control IgG), $n = 7$ (treated). Statistical significance (CDX3379-MMAE + IR vs. CDX3379-MMAE + IR + anti-PD-1) calculated using Log-rank (Mantel–Cox) test. Source data in Source Data file.

difficult to control with the combination of MMAE + IR alone (Fig. 5c, left panel). However, the addition of anti-PD-1 therapy to MMAE + IR improved tumor control of these larger MC38 tumors. In addition, we evaluated the trimodal combination of MMAE, IR, and anti-PD-1 therapy in orthotopically grown 4MOSC1 tumors (Fig. 5d). Again, tumor regression was superior when immune checkpoint inhibition was delivered together with MMAE + IR. Finally, we tested if anti-PD-1 therapy potentiated MMAE radiosensitization irrespective of the coupled carrier vehicle delivering MMAE, i.e., cell penetrating peptide (ACPP) or antibody (ADC) (Fig. 5e, f). Importantly, trimodal ACPP or ADC tumor-targeted MMAE in combination with IR and anti-PD-1 antibody achieved statistically significant durable tumor control independent of drug delivery platform when compared to ACPP-MMAE + IR or CDX3379-MMAE + IR. Tri-modality therapy with MMAE delivered by as an ADC or ACPP conjugate was well tolerated as measured by mouse body weight (Supplementary Fig. 9f). Taken together, these results demonstrate improved tumor control when immune checkpoint inhibition is integrated with MMAE radiosensitization and that anti-PD-1 immune checkpoint potentiation of MMAE radiosensitized tumor kill is intrinsic to the actions of MMAE since it is not limited by the specific carrier vehicle to which MMAE is attached.

## Discussion
Based on these findings, we propose the therapeutic index of the concurrent chemo-radiotherapy paradigm can be increased by leveraging antibody drug conjugation to spatially target highly potent radiosensitizing auristatins specifically to irradiated tumors while simultaneously avoiding normal tissue. Locally advanced non-metastatic cancers continue to pose therapeutic challenges in the era of precision medicine and immunotherapy. To test a molecularly targeted chemo-radio-immunotherapy strategy, we synthesized complementary tumor-targeted antibody and peptide auristatin conjugates and investigated their interactions with IR in the context of an intact immune system. For proof-of-concept, we focused on auristatin radiosensitization since MMAE is the most common ADC warhead in clinical development[22]. Combining MMAE with IR maximizes upfront tumor kill by independent mechanisms of action. In addition to MMAE's intrinsic tumoricidal activity, MMAE has the added advantage of sensitizing tumor cells to IR-induced DNA damage. Finally, tumor-cell-specific MMAE radiosensitization sculpts the tumor immune microenvironment promoting anti-tumor immune responses and immunologic memory that augment immune checkpoint inhibition therapy. A strength of the approach taken in the current studies is that we have shown MMAE warheads improve radiosensitized immunogenic tumor

control irrespective of how MMAE is delivered to tumors (i.e., ADC or ACPP conjugates), suggesting these results are broadly applicable to the growing pharmaceutical pipeline of auristatin armed ADCs and will yield similar therapeutic synergies when integrated with radiotherapy and immunologics.

Radiosensitization approaches targeting ErbB signaling have exclusively centered on signal inhibition but have proven inferior to concurrent cytotoxic chemotherapies in clinical trials. While there is strong rationale for receptor signaling inhibition of cancer cells, blocking receptor tyrosine kinase signaling invariably activates bypass pathways resulting in resistant clones. This creates a whack a mole problem which is evidenced by the clinical experience[42,58]. ADCs overcome this issue by using signaling receptors as localization beacons for targeted delivery of tumor-icidal drugs. Auristatins are highly potent cytotoxins and effective across genetically diverse tumor cell lines and auristatin ADCs routinely induce human tumor xenograft regression[39,59–62]. To this point and in agreement with prior ADC xenograft studies, we demonstrated that CDX3379-MMAE ADC induced tumor xenograft regression. Importantly, isogeneic human CAL27 cell lines with differential sensitivity to anti-HER3 receptor blocking antibody both showed exquisitely sensitive to antibody-conjugated MMAE (Fig. 2c). Somewhat surprising when we advanced studies to syngeneic models, MMAE monotherapy produced only modest tumor control. Across a panel of diverse cancer models, including routinely used LL/2, MC38 and B16 tumors, tumors invariably grew out when treated with MMAE alone (Figs. 2, 3). Moreover, MMAE monotherapy lacked therapeutic efficacy irrespective of the coupled tumor-directed delivery vehicle, i.e., antibody (i.e., CDX3379) or cell penetrating peptide (i.e., ACPP), implying that MMAE's ineffectiveness was not a limitation of tumor cell receptor expression.

Given that auristatins show radiosensitizing potential, we hypothesized concurrently given tumor-targeted MMAE and IR would synergize cytotoxic tumor kill and engage anti-tumor immune responses. This has translational implications for two key clinical oncologic issues, i.e., therapy resistance and treatment induced side effects. First, recent reports have demonstrated tumor resistance to ADCs develop which limits the window of opportunity for ADCs[63–65]. Utilizing auristatin's dual properties of cytotoxicity and radiosensitization has the advantage of maximal upfront tumor cell kill which minimizes opportunities for therapy resistant clonogens to emerge. Second, combinatorial therapies require reduced dose of each agent to achieve similar levels of tumor control that can translate into improve patient tolerability of therapies. Clinical experience with ADCs has revealed systemic toxicities suggesting a narrow therapeutic index even with such molecularly precise therapies[64,66–68]. Utilizing

ADCs at a lower radiosensitizing doses has the potential to maintain tumor control while decreasing debilitating toxicities of ADC warheads. Alternatively, the MMAF auristatin derivative has decreased bystander effects which could decrease bystander peri-tumoral radiosensitization after being released from the ADC in receptor positive tumor cells[39]. When we tested CDX3379 armed with auristatin derivatives with IR, CDX3379-MMAE had improved tumor control compared to CDX3379-MMAF suggesting bystander effects are advantageous to auristatin radiosensitization in immune-competent models.

In the last decade, therapeutics targeting immune checkpoint inhibitors (i.e., CTLA-4, PD-1, and PD-L1) have revolutionized cancer care for subsets of oncology patients[24,25]. There is tremendous interest in developing strategies to further improve immune checkpoint inhibitor responses to wider patient populations. To achieve increasing therapeutic precision, integrating the specificity of ADC with inherent targeting of radiotherapy and immunotherapy is appealing. Interestingly, two recent studies have begun to evaluate ADCs and immunologics. U3-1402 is a HER3-targeted ADC with a topoisomerase I inhibitor (DXd) warhead that sensitized tumors to PD-1 blockade[69]. Enapotamab vedotin is an AXL-targeted ADC conjugate with an MMAE warhead that also enhanced tumor immunity[45]. In melanoma and lung cancer models enapotamab vedotin-induced immunogenic cell death and prolonged mouse survival. However, while enapotamab vedotin prolonged tumor control, it failed to generate durable tumor control suggesting a role for cytotoxic therapies with alternative mechanisms of tumor kill. Importantly, both of these studies were done without radiotherapy, a treatment used in over 50% of cancer patients. Since ionizing radiation is inherently cytotoxic and engages anti-tumor immune responses, radiotherapy is actively being investigated in clinical trials with immunotherapies[27,29,31–35]. In pre-clinical studies, IR doses have primarily been larger single (i.e., 20 Gy × 1) or fractionated (i.e., 8 Gy × 3) regimens that ablate the irradiated tumor target to generate systemic anti-tumor immune responses, i.e., abscopal effects[26,70]. Our approach was conceptually different in that we attempted to leverage the highly successful concurrent chemo-radiotherapy paradigm by integrating immunotherapies with precision radiosensitization. The fractionated IR dose regimens (i.e., 3–5 Gy × 3) we employed produced modest growth delay in irradiated tumors as monotherapy with or without anti-PD-1 antibody. Importantly, MMAE combined with these smaller fractioned IR doses results in CD8 T cell-dependent durable tumor control, induces immunologic memory and potentiates anti-PD-1 immune checkpoint inhibition.

The concurrent chemo-radiotherapy paradigm resulted in a major leap forward in treating locally advanced cancers. Since its advent, technological advancements in radiotherapy delivery using advanced radiographic imaging inputs (including PET tracers and MRI sequences) have allowed for IR to be delivered with increasing conformality to tumors using intensity modulation (IMRT) and image guidance (IGRT) techniques[10]. However, molecularly guided systemic therapies incorporated into the chemo-radiotherapy paradigm have been lacking in the era of precision medicine[32]. Taken together, our findings suggest that improvements to the concurrent radiotherapy-systemic therapy paradigm may be achieved by integrating tumor-directed radiosensitizing auristatins with radiotherapy and immunotherapy. From a translational perspective, it will be of interest to test the growing number of auristatin ADCs as well as other classes of ADC warheads for their ability sensitize the irradiated tumor immune microenvironment[67]. In summary, these studies lay a foundation for improving upon conventional non-targeted chemo-radiotherapy toward spatially precise, biomarker-driven radio-chemo-immunotherapies.

## Methods

**Cells and reagents**. Human head and neck cancer cell lines (CAL27 (CRL2095), Detroit 562 (CCL-138)), murine lung (LL/2 (CRL-1642)) and murine melanoma (B16 (CRL-6475)) were obtained from American Type Culture Collection. Murine colorectal (MC38 (ENH204-FP)) cancer cell lines were obtained from Kerafast. Murine head and neck MOC1 and MOC2 were generously provided by Dr. R. Uppaluri. CAL27 PIK3CA H1047R expressing cells and 4MOSC1 cells were generated in Dr. Gutkind's laboratory and previously characterized[42,55]. CAL27, Detroit 562, LL/2, and B16 cells were cultured in DMEM (Gibco) supplemented with 10% FBS (Omega Scientific). MC38 cells were cultured in DMEM supplemented with 10% FBS, 1 mM sodium pyruvate (Gibco), 1% non-essential amino acids (Gibco), and 10 mM HEPES (Gibco). MOC1 and MOC2 cells were cultured in IMDM (Lonza)/F-12K nutrient mixture at a 2:1 mixture with 5% FBS, 5 ng/ml EGF (EMD Millipore), 400 ng/ml hydrocortisone (Sigma), and 5 mg/ml insulin (Sigma). On receipt, each cell line was expanded, cryopreserved as low passage stocks, and routinely tested for mycoplasma. Cisplatin (Enzo Biosciences), paclitaxel (Sigma), and MMAE (Concortis) were reconstituted in DMSO. Clinical-grade CDX3379 was provided by Celldex. Clinical grade cetuximab (Erbitux, Lilly Medical) was obtained from UCSD Moores Cancer Center Pharmacy.

**Synthesis of Cy5-labeled antibody-conjugated MMAE and MMAF**. Anti-HER3 CDX3379 antibody MMAE or MMAF drug conjugate was synthesized using methods previously described[39]. A solution (82 nmol, 0.25 ml, 50 mg/ml) of CDX-3379 was mixed with sodium bicine buffer (50 µl, 1 M pH 8.3), sodium diethylenetriaminepentaacetic acid (10 µl, 100 mM pH 7) and water (0.69 ml). Following reduction with 4 equivalents of tris(carboxyethyl)phosphine (TCEP; 327 nmol, 32.7 µl of 10 mM solution in water) at 37 °C for 2 h, the solution was added to 4 equivalents of maleimidocaproyl-valine-citrulline-PABC-MMAE (MC-VC-MMAE, Levena Biopharma; 327 nmol, 32.7 µl of 10 mM solution in DMSO; prediluted with 32.7 µl of water immediately before addition) or 4 equivalents of maleimidocaproyl-valine-citrulline-PABC-MMAF (MC-VC-MMAF, Levena Biopharma; treated as for MMAE linker). After 30 min at room temperature, 2 equivalents of Cy5-maleimide (164 nmol, 16.7 µl of 10 mM solution in DMSO; prediluted with 16.7 µl of water immediately before addition) was added and after a further 30 min, gel-filtered (Sephadex G25, 1.0 g) eluting with PBS. Following centrifugal concentration (Centricon 30 kDa MWCO) to ~500 µl, the concentrations of antibody and Cy5 were determined by absorbance using extinction coefficients of 225,750 M⁻¹ cm⁻¹ (CDX3379) at 280 nm and 12500 M⁻¹ cm⁻¹ and 250,000 M⁻¹ cm⁻¹ at 280 and 650 nm, respectively, for Cy5. The solution was diluted with PBS to a final antibody concentration of 100 µM, yielding 77 nmol. Drug loading was measured by denaturing reverse-phase HPLC (Agilent PLRP-S column at 90 °C; acetonitrile–water–0.05% TFA gradient) of the reaction mix prior to addition of Cy5 maleimide, following reduction of any remaining intersubunit disulfides with 50 mM DTT for 30 min. Peaks corresponding to light or heavy chains with 0–3 MMAE or MMAF were identified by electro-spray mass spectroscopy (Supplementary Fig. 2) and peak areas at 280 nm were integrated and weighted to calculate the drug loading. Modified light chain (L1) and unmodified H chain (H0) were not resolved so MMAE and MMAF loading is an underestimate. No free MC-VC-MMAE or MC-VC-MMAF was detected by HPLC following gel filtration. Cetuximab-conjugated MMAE was synthesized as previously described[37].

**Synthesis of ACPP-MMAE**. Base ACPP with H₂N-peg8-e₉-oPLGC(Me)AG-r₉-c-CONH₂ was made using regular solid phase Fmoc peptide synthesis (Supplementary Fig. 5), where lower case letters refer to D-amino acids, peg8 refers to H₂N-PEG8-propionic acid, o-denotes for 5-amino-3-oxopentanoyl a short hydrophilic spacer, C(Me) denotes for S-methylcysteine and the final CONH₂ indicates C-terminal amide. The peptide was isolated from the resin by treating it with mixtures containing 92% trifluoroacetic acid (TFA), 2% thioanisole, 2% water, and 4% triisopropylsilane (TIPS) for 4 h under N₂ atmosphere and filtered. This filtrate was concentrated and then precipitated by addition of ice cold 50% hexanes in ethyl acetate mixture. Centrifugation was performed to isolate the precipitate that was dried under vacuum. The peptide was dissolved in DMSO and purified by HPLC using 5–55% acetonitrile in water and 0.05% TFA over a period of 25 min at 3.5 ml/min flow rate. The purified product was dried using lyophilization. ES-MS found 621.4 (M⁺ + 6H⁺), 745.8 (M⁺ + 5H⁺), 932.0 (M⁺ + 4H⁺), 1242.4 (M⁺ + 3H⁺), deconvolved to 3724.9 (M⁺ + H⁺), calculated for $C_{147}H_{259}N_{55}O_{54}S_2$, 3723.87. H₂N-peg8-e₉-oPLGC(Me) AG-r₉-c-CONH₂ (10·TFA salt, 9.90 mg, 2.0 µmol) dissolved in dry DMSO (200 µl) was added to MC-VC-PABC-MMAE (2.70 mg, 2.0 µmol) and N-methyl morpholine (2.2 µl, 20 µmol) was added with mixing. LC–MS indicated complete reaction after 30 min to give a single product, ES-MS found 841.2 (M⁺ + 6H⁺), 1009.2 (M⁺ + 5H⁺), 1261.2 (M⁺ + 4H⁺), 1681.5 (M⁺ + 3H⁺), deconvolved to 5041.0, calculated for $C_{215}H_{364}N_{66}O_{69}S_2$, 5041.8 that was used without further purification. A solution of 6-maleimidocaproic acid N-hydroxysuccinimide ester (Sigma-Aldrich; 20 µl of 100 mM in dry DMSO, 2.0 µmol) was added to the reaction mixture and kept at room temperature for 4 days until LC–MS showed reaction was complete. ES-MS found 1049.5 (M⁺ + 5H⁺), 1309.5 (M⁺ + 4H⁺), 1745.1 (M⁺ + 3H⁺), deconvolved to 5233.8, calculated for $C_{225}H_{375}N_{67}O_{72}S_2$, 5235.0. Cyclo(RGD)fC (Peptides International, 1.5 mg, 2.6 µmol) dissolved in dry DMSO (100 µl) was added and mixed. LC–MS indicated complete reaction after 30 min to yield final product, and the reaction was quenched with acetic acid

(50 μl), separated by HPLC and lyophilized to give cyclo(RGD)fc-MC-HN-peg8-e₉-oPLGC(Me)AG-r₉-c-(MC-VC-PABC-MMAE)-CONH₂ as a white powder, yield, 7.67 mg (56%). ES-MS found 831.4 ($M^+ + 7H^+$), 969.8 ($M^+ + 6H^+$), 1163.5 ($M^+ + 5H^+$), deconvoluted to 5812.5, calculated for $C_{249}H_{409}N_{75}O_{79}S_3$, 5813.6.

**Synthesis of ratiometric and Cy5-labeled ACPPs.** Ratiometric activatable cell-penetrating peptide was synthesized as previously described (Supplementary Fig. 6a)[48]. For Cy5-labeled ACPP conjugate, 1.1 equivalent of Cy5-maleimide (MW = 778 g/mol) was coupled to the ACPP peptide, followed by acylation with cyclo(RGDfC) as described for ACPP-MMAE conjugate (Supplementary Fig. 6b). Peptides was lyophilized and stored as a powder at −20 °C.

**Cytotoxicity assays.** Cells were plated in 96-well plates and exposed to a drug concentration range for 72 h. Alamar Blue was added to the cells and allowed to incubate for 2–4 h at 37 °C. Plates were analyzed using a plate reader with fluorescence measured at 560 nm. Fractional survival was normalized to untreated control cell fluorescent values. Drug $IC_{50}$ was calculated using Prism software (GraphPad, La Jolla, CA). Cell viability, cytotoxicity and apoptosis were also measured by ApoTox-Glo Triplex Assay (Promega).

**Immunoblotting.** Cells were harvested and lysed in RIPA buffer (20 mM Tris pH 8, 50 mM NaCl, 15 mM EDTA, 1% Triton X-100) with protease and phosphatase inhibitors (Complete Protease Inhibitor Cocktail and Phos-Stop, Roche). Protein lysate was quantitated by the BCA technique (Pierce). 20 μg of lysate underwent electrophoresis using 4–12% Bis–Tris gels (Life Technologies), transferred to PVDF membranes (iBlot) and incubated with indicated primary antibodies HER3, GAPDH (Cell Signaling Technology, Cat# 12708 and Cat# 2118) at dilutions of 1:2000 and 1:5000, respectively. Blots were developed by ECL (Pierce). Uncropped blots are shown in Source Data file.

**Cell cycle.** Cells were MMAE treated for 24 h, fixed in 70% ethanol, stained with FxCyclePI/RNase solution (Life Technologies), and then analyzed by FACS using FlowJo software.

**Clonogenic assay.** Cells were treated with MMAE for 24 h and then irradiated with 2 Gy. For cell irradiation, IR was delivered using a PXI X-RAD 225 XL irradiator at maximal dose rate of 1.87 Gy/min with a beam conditioning copper filter (Precision X-ray Irradiation). Following IR, cells were counted, re-plated at varying cell numbers in drug free media. 10–14 days after initial seeding formed colonies were methanol fixed, stained with crystal violet, and counted. Surviving fraction at 2 Gy (SF2) was calculated as the fraction of cells surviving 2 Gy compared to non-irradiated cells.

**In vivo Cy5 ADC optical imaging.** All animal work was done in compliance with the University of California San Diego Institutional Animal Use and Care Committee under protocol #S15290. Mice were housed in individually ventilated and micro-isolator cages supplied with acidified water and fed 5053 irradiated PicoLab Rodent Diet 20. Temperature for laboratory mice in housing was 18–23 °C with humidity 40–60%. Housing room is maintained on a 12-h light/dark cycle. For human tumor xenografts, 4–6-week-old female athymic nu/nu mice (UC San Diego Animal Care Program breeding colony) were injected subcutaneously into the bilateral thighs with $5 \times 10^6$ CAL27 or Detroit 562 tumor cells in a 1:1 Matrigel (BD) and PBS solution. For syngeneic subcutaneous tumors, 6-week-old female C57BL/6 albino mice (Jackson Labs) were injected subcutaneously into the bilateral thighs with MOC1, MOC2 or LL/2 tumor cells. For orthotopic tumors, 6-week-old female C57BL/6 mice (Jackson Labs) were injected into the tip of the tongue with MOC2 tumor cells. Mice were intravenously (i.v.) injected with Cy5-labeled CDX3379-MMAE when subcutaneous tumors were >100 mm³ or on day 7 post implant for orthotopic tumors. For imaging, mice were anesthetized (1:1 mixture of 100 mg/ml of ketamine and 5 mg/ml of midazolam). Animals were imaged using a Maestro Small Animal Imager (CRI) with excitation filter of 607/36 and 633 nm long pass emission filter with dichroic filter tuned to 670 nm for Cy5 images only. Imaging was done both with skin on and after skin removal to decrease auto-fluorescence and scattering. Exposure time for Cy5 signal capture for tumor models was as follows: CAL27 for 800 ms, MOC1 for 200 ms and MOC2 for 1500 ms. Tissues were harvested from CAL27 tumor-bearing mice, sectioned and stained with guinea pig anti-cytokeratin 5 (CK5) (Fitzgerald, MA, USA), followed by anti-guinea pig Alexa 488 to demarcate tumor and adjacent normal muscle tissue. CK5 images were taken using the 488 nm laser line, Cy5 fluorescence images were taken using the 640 nm laser line and DAPI nuclear stain were imaged using the 405 nm laser line on a Nikon A1 confocal microscope with ×20/0.75 NA objective (tumor and peri-tumoral muscle tissues) acquired at high resolution using Nyquist function (0.2 μm/px). Cy5 fluorescence for resected LL/2 and MC38 tumors were measured using IVIS 200 (Xenogen). For blood clearance studies, the tail was pricked at indicated time points after injection, and a small volume (5–10 μl) of blood was collected in a heparinized hematocrit tube. Fluorescent images were taken using the Maestro with filters mentioned above with 200 ms exposure time and integrated fluorescent intensity quantitated using Image J software.

**In vivo Cy5 and Cy7 ACPP optical imaging.** LL/2 or MC38 tumor cells were subcutaneously implanted into both thighs of 6-week-old female C57BL/6 albino mice (Jackson Labs). 4MOSC1 tumor cells (150,000) were orthotopically injected in tongues of 6-week-old female C57BL/6 mice. When tumors were palpable, mice were anesthetized (1:1 mixture of 100 mg/ml of ketamine and 5 mg/ml of midazolam) and i.v. injected through the tail vein with 10 nmoles of Cy5 and Cy7-labeled ratiometric ACPP or Cy5-labeled ACPP in 100 μl of sterile water. At 90 min after injection, mice were sacrificed and whole mouse imaged using the Maestro small animal imager (CRI). Imaging was done both with skin on and after skin removal to decrease autofluorescence and scattering. Tissues were then harvested and imaged individually for Cy5 and Cy7. The acquisition parameters were excitation filter 607/36 scanned across a wavelength of 640–840 nm, emission filter 633 LP. The in vivo images were analyzed with custom software for generating Cy5/Cy7 ratiometric pseudocolor images. Exposure time for fluorescence capture for tumor models was as follows: LL/2 for 800 ms, MC38 for 300 ms. For 4MOSC1 tumors injected with ACPP-Cy5 the exposure time for resected tongues was 300 ms at 670 nm dichroic. These tongues were then sectioned and confocal imaged with a ×4/0.20 NA objective. White light mouse images were captured with a 3–5 ms exposure time.

**Gelatin zymography assays.** 6-week-old female C57BL/6 albino mice (Jackson Labs) were injected subcutaneously with B16, MC38, or LL/2 cells subcutaneously into the hindlimbs in 100 μl of 1:1 Matrigel (BD) and PBS solution. When tumors were palpable, tumor, and peri-tumoral muscle were harvested and frozen in liquid nitrogen. Samples were placed in cold NP-40 lysis buffer containing both phosphatase and protease inhibitors, homogenized by passing through a syringe, centrifuged at 16,000×g for 10 min at 4 °C, and protein concentration measured on collected supernatants. 2 × Tris glycine SDS sample buffer was added to the samples and equivalent amounts of total protein loaded onto 10% zymogram gelatin gels (ThermoFisher). Gels were developed using Novex zymogram renaturing and developing buffers (10×) and stained with SimplyBlue Safestain (ThermoFisher). Purified MMP-2 and MMP-9 controls were loaded and run to identify standard locations of gelatinase activity. Uncropped blot is shown in Source Data file.

**Tumor drug measurement.** For CDX3379-MMAE drug measurements, 6-week-old female NOD SCID (University of California San Diego Animal Care Program) were injected orthotopically into the tongue with 150,000 CAL27 PIK3CA H1047R expressing cells in 20 μl of DMEM. After 10 days, mice were intravenously (i.v.) injected through the tail vein with 2.5 nmole of CDX3379-MMAE in 50 μl of PBS. For ACPP-MMAE tissue drug measurements, 6-week-old female C57BL/6 albino mice (Jackson Labs) were injected subcutaneously with 50,000 B16 or LL/2 cells subcutaneously into the hindlimbs in 100 μl of 1:1 Matrigel (BD) and PBS solution. When tumors were palpable, mice were i.v. injected through the tail vein with 10 nmoles of ACPP-MMAE in 100 μl of sterile water. Tumors were excised on days indicated in figure legends, weighed, and homogenized in 10 v/w of 2% acetic acid in 1:1 acetonitrile:water with a point sonicator (Fisher Scientific) using an amplitude range of 5–15% for 20 s on ice. Homogenates were centrifuged (14 × g, 10 min), supernatants collected for drug concentration measurements using LC–MS/MS. Drug concentration was determined by LC–MS/MS with Luna-2 C18 column and Agilent Trap XCT mass spectrometer. For MMAE tissue concentrations, integrated ion currents for m/z 682.4, 496.4 and m/z 686.4, 506.4 extracted from fragmented m/z 740.4 and 718.4, respectively, were combined. The total ion current was fit to a standard curve generated for each drug to quantitate tissue drug concentration.

**In vivo orthotopic tumor therapy efficacy experiments.** Orthotopic tongue xenograft tumors were established in 6-week-old NOD SCID mice by injecting 150,000 CAL27 parental (WT) or PIK3CA H1047R-expressing cells. Once tumors were palpable, mice were i.v. injected with ACPP-MMAE, CDX3379 or CDX3379-MMAE conjugate as indicated in figure legends. Maximal tumor volume approved on our protocol for orthotopic oral cavity tumors was 150 mm³. On the last day of tumor xenograft measurements, mice were euthanized, tongues resected, imaged, and fixed. For in vivo orthotopic syngeneic tumor ACPP therapy studies, 6-week-old female C57BL/6 mice (Jackson Labs) were injected with 150,000 4MOSC1 cells into the tongue or buccal mucosa. Five days after implanted mice were treated with either ACPP, ACPP-MMAE, and IR as indicated in figure legends. For orthotopic tongue tumor irradiation, tumors targets were delineated and given 2 Gy using PXI SmART irradiator with 225 kVp X-ray tube. IR was delivered as 2 parallel opposed fields. Tumor volumes were measured and calculated using the formula as ½ * length * width².

**Orthotopic tumor CyTOF analysis.** 6-week-old C57BL/6 female mice were implanted with 50,000 MOC2 cells into the tongue. After tumors reached ~20 mm³, mice were i.v. injected with 2.5 nmoles of CDX3379 or CDX3379-MMAE. Five days after treatment, tumors were isolated, minced, and re-suspended in FBS-free DMEM media supplemented with components of MACs tumor dissociation kit. Tissues were incubated for 15 min at 37 °C and mechanically digested using the gentle MACs Octo Dissociator. Digested samples were then passed

through a 100-μm strainer to acquire a single-cell. For viability staining, cells were washed in PBS and stained with Cell-ID Cisplatin (DVS Sciences) to a final concentration of 5 μM for 5 min at room temperature. Cisplatin was quenched when cells were washed and stained with the antibody cocktail. Antibodies were prepared in Maxpar cell staining buffer (PBS with 2 mM EDTA, 0.1% BSA, 0.05% NaN$_3$) and incubated with cells for 15 min at room temperature. After staining, cells were washed and fixed with 1.6% formaldehyde (FA) for 10 min at room temperature. For cell identification, cells were washed in staining buffers and stained with DNA intercalator (Fluidigm) containing natural abundance Iridium (191Ir and 193Ir) prepared to a final concentration of 125 nM. Antibodies (clone in parenthesis) used for CyTOF included B220 (RA3-6B2), CD117(2B8), CD11c (N418), CD25 (3C7), CD4 (RM4-5), CD45 (30-F11), CD8a (53-6.7), MHC-II (M5/114.15.2), NKP46 (29A1.4), TCRb (H57-597) from Fluidigm. Antibodies CD103 (2E7), CD115 (AFS98), CD11b (M1/70), CD19 (6D5), CD23 (B3B4), CD3 (145-2C11), CD64 (X54-5/7.1), F4/80 (BM8), FCeRI (Mar-1), FR4 (TH6), Ly6C (HK1.4), Ly6G (1A8), NK1.1 (PK136) from BioLegend and antibody Siglec-F (1RNM44N) from ebioscience. All antibodies were validated by the supplier and used at a 1:100 dilution. Cells were washed with staining buffer and pelleted. Before acquiring, cells were resuspended in 0.1× dilution of EQ Four Element Calibration beads (Fluidigm) and filtered through a 35 μm nylon mesh filter. Cells were acquired on a Helios CyTOF Mass Cytometer (Fluidigm) at an event rate of 200 events/s or less. Data was normalized using Matlab-based normalization software based on the EQ bead-removal. Prior to analysis, samples were subsampled to the lowest number of events and all counts were normalized to the same number of total events, 2324. To detect clusters of cells with a similar expression of surface markers in CyTOF, single cells were gated and clustered using unsupervised dimensionality reduction algorithm optimal t-Distributed Stochastic Neighbor Embedding (opt-SNE) algorithm in OMIQ data analysis software (www.omiq.ai) (530 iterations, Perplexity 30, and Theta 0.5)[51].

**In vivo subcutaneous tumor therapy experiments.** MOC1, B16, LL/2, and MC38 tumor cells were injected subcutaneously into the hindlimbs of 6-week-old female C57BL/6 albino mice in 100 μl of 1:1 Matrigel (BD) and PBS solution. Tumors were injected with CDX3379, CDX3379-MMAE, ACPP, or ACPP-MMAE as indicated in figure legends. Fractionated IR was delivered focally to the tumor-bearing hindlimbs (3 Gy to MC38 tumors and 5 Gy to LL/2 and B16 tumors) as indicated in figure legends using a PXI X-RAD 225 XL irradiator at maximal dose rate of 3 Gy/min with a beam-conditioning aluminum filter (Precision X-ray Irradiation). For RAG1 KO studies, 6-week-old female Rag1$^{tm1Mom}$ mice were injected subcutaneously with tumor cells in their hindlimbs (Jackson Labs). For CD8 depletion studies, mice were i.p. injected with anti-CD8 (BioXCell #BE0117) at 10 mg/kg on days −6, −5 and −4 tumor cell implant and then weekly starting on day after tumor cells were injected or control antibody. For anti-PD-1 therapy, mice were i.p. injected with 200 μg anti-PD-1 antibody (BioXCell #BE0033) or control antibody on days 3, 6, and 11 after tumor cell implantation. For immunologic memory studies, MC38 mice with no residual tumors following treatment were then challenged with re-injection of MC38 tumor cells as indicated in figure legends. To ensure tumor cell viability on day of re-injection, a set of tumor naïve mice were also injected with MC38 tumor cells on day of challenge re-injection. Tumor volumes were measured and calculated using the formula as ½ * length * width$^2$. Mice were sacrificed when tumor volume reached >1500 mm$^3$, maximal tumor volume approved on our protocol for subcutaneous tumors.

**4NQO carcinogen-induced tumor model.** 4-Nitroquinoline N-oxide (4NQO, Sigma Aldrich-N8141) was dissolved in propylene glycol as a stock solution (4 mg/ml) and further diluted in drinking water to a final concentration of 50 μg/ml. 6-week-old female C57BL/6 mice were exposed to 4NQO-laced water for 16 weeks (refreshed weekly) after which all animal cages were reverted to plain water[55]. At week 20, mice were either treated with ACPP-MMAE and IR as indicated in figure legends. Since the entire tongue was exposed to 4NQO, the whole tongue was delineated and given 5 Gy × 2 using PXI SmART irradiator with 225 kVp X-ray tube. IR was delivered as 2 parallel opposed fields. All mice were given a full oral cavity examination biweekly, and any observed pathological changes were documented. On week 22, animals were euthanized, and tissues and oral pharynx were fixed in PAXgene tissue FIX (Qiagen) at room temperature for 12 h. Tissues were then transferred to 70% ethanol and paraffin embedded. Tongue sections were H&E stained and histopathologically assessed for the number and size of carcinoma lesions per tongue by a qualified Pathologist (A.A.M.) who was blinded manner to treatment of each slide.

**NanoString and qPCR analysis.** RNA was isolated from cells or tumor tissues and comprehensive immune profiling was analyzed using the NanoString PanCancer IO 360 Panel and NanoString nCounter PanCancer Mouse Immune Profiling gene expression platforms. Gene set analysis with calculated global significance scores and directed significance scores were calculated for treated cells. The Advanced Analysis module of the nSolver software was used to analyze genes associated with listed immune cells in tumors and given a Z-score ($n = 3$ mice per group). Expression levels of GZMB were analyzed using real-time quantitative PCR.

Reverse transcription was performed with the iScript cDNA synthesis kit (Biorad), using aliquots of total RNA extracted with RNeasy Plus Mini Kit (Qiagen). Gene-specific primers were GZMB-F TCTCGACCCTACATGGCCTTA and GZMB-R TCTCGACCCTACATGGCCTTA. Real-time PCR reactions were performed using the CFX Connect Real-Time PCR System (Biorad) and the amplifications were done using the iTaq Universal SYBR Green Supermix (Biorad). qPCR data were analyzed using the Bio-Rad CFX Maestro software.

**Flow cytometry analysis.** B16 tumors were established as above in 6-week-old female C57BL/6 albino mice. Mice were then treated with MMAE and IR and harvested as indicated in figure legends. At harvest, tumors were isolated, minced, and re-suspended in FBS-free DMEM media supplemented with components of MACs tumor dissociation kit. Tissues were incubated for 15 min at 37 °C and mechanically digested using the gentle MACs Octo dissociator. Tissue suspensions were washed with fresh media and passed through a 100-μm strainer. Samples were split in half for immediate staining or stimulated with the Biolegend Cell Activation Cocktail (with Brefeldin A) for 3 h. Samples were washed with PBS and immediately processed for live/dead cell discrimination using BD LIVE/DEAD™ Fixable Blue Dead Cell Stain Kit. Cells were washed and stained for surface markers for 30 min at 4 °C. Intracellular staining was performed using the eBioscience FOXP3/Transcription Factor Buffer Set from Invitrogen and stained with intracellular antibodies. Antibodies (clone and dilution in parenthesis) were purchased from BioLegend and included CD45 (30-F11, 1:100), CD90.2 (30-H12, 1:200), CD8a (53-6.7, 1:100), CD4 (RM4-4, 1:400), NK1.1 (PK136, 1:100), PD-1 (29F.1A12, 1:100). All antibodies were validated by the supplier. All flow cytometry data acquisition was done using BD LSRFortessa and analyzed using FlowJo software. Immune cells were identified by the following characteristics: cytotoxic T cells (CD45+Thy1.2+CD8+), helper T cells (CD45+Thy1.2+CD4+), NK cells (CD45+Thy1.2−NK1.1+).

**Statistical analysis.** Unpaired two-sided $t$ tests were performed for quantitative Cy5:Cy7 ratio, tissue drug concentrations, cell culture clonogenic survival, flow cytometry and end of study tumor volume responses. For temporal tumor volumetric response studies, two-way ANOVA analysis was performed with Tukey's multiple comparison group. Survival curves were analyzed Log-rank. All statistical analyses were performed using Prism software (GraphPad).

**Reporting summary.** Further information on research design is available in the Nature Research Reporting Summary linked to this article.

## Data availability
All data reported in this work are available within the Article, Supplementary Information, Supplementary Data, or Source Data file. Source data are provided with this paper.

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

## Acknowledgements

This work was supported by the National Institutes of Health/National Cancer Institute CA215081 (S.J.A.), CA219744 (S.J.A.), P30CA23100 (Microscopy Core, Moores Cancer Center), S10OD018499 (Flow Cytometry Core Facility, La Jolla Institute), CIRM Major Facilities grant FA1-00607 (UC San Diego Stem Cell Genomics Core, Sanford Consortium for Regenerative Medicine), and University of California Cancer Research Coordinating Committee CRR-17-422691 (S.J.A.). The authors thank Kersi Pestonja-masp for technical microscopy assistance and Dr. Elsa Molina for technical assistance with NanoString studies. Mouse image in Fig. 3g was created with Bio.Render.com (License Agreement# QO242QCDWR).

## Author contributions

D.V.H., M.M.A., and S.J.A. conceived and designed the studies. D.V.H. and S.R.A. synthesized the antibody and peptide drug conjugates. D.J.S. calibrated radiation therapy X-ray source. Z.W., D.A., and J.S.G. provided reagents. D.V.H., M.M.A., M.F.C., J.L., M.A.Q., J.A., and I.F.P. performed the experiments. D.V.H., M.M.A., A.A.M., A.B.S., J.D.B., E.E.W.C., S.R.A., J.S.G., and S.J.A. interpreted the data. S.J.A. supervised the studies.

## Competing interests

University of California San Diego has filed patent applications based on the findings described in this manuscript: "Tumor radiosensitization with monomethyl auristatin E (MMAE) and derivatives thereof" USPTO #10596259 (S.J.A.), "Tumor radiosensitization with antibody conjugates" US Patent Application No. 15/817,017 (S.R.A. and S.J.A.). The remaining authors declare no competing interests.
