## [Peer Review File · Nature Communications]

Monomethyl auristatin antibody and peptide drug conjugates for trimodal cancer chemo-radio-immunotherapyREVIEWER COMMENTS

Reviewer #1 (Remarks to the Author): with expertise in antibody drug conjugates, cancer therapy

Dear editor,

In this work by Hingorani et al authors explore the potential of the trimodal precision chemio-radio-immunotherapy using a combination of MMAE loaded vehicles, IR and clinical approved anti-PD1 mAb.

Although the finding of this work are significant, especially from a clinical translation perspective, how the entire manuscript has been set is not entirely convincing.

In the first part of the paper, Figure 1,2 and 3 authors show data about activity of an anti-HER3 ADC or activable cell penetrating peptide (ACPP) coupled with MMAF alone and/or in combination with IR. These data are just a repetition of what the same group already presented in two recent papers : 1) 2016 (Adams et al, NatComm DOI: 10.1038/ncomms13019) where, using other ErbB receptor as targets (i.e. HER1 and HER2) it was consistently demonstrated that MMAE based ADCs are endowed with the ability to restrict payload radiosensitization to the tumor cells/tissue. 2) 2020 (Hingorani et al, Biomaterials <https://doi.org/10.1016/j.biomaterials.2020.120032>), where it's described the advantages of protease triggered cell penetrating peptide scaffolds to efficiently deliver radiosensitizing payloads (such as MMAE) increasing efficacy of radiotherapy. In summary, the data presented in the first part of this paper are not novel and possibly out of the scope of this paper.

Indeed, the novelty here is represented by the mechanistic definition of how coupled MMAE is able to control tumor growth by radiosensitization of cancer cells. In this aspect authors reveal that payload radiosensitization and tumor control is mediated by adaptative immunity. In particular, they demonstrated that the combination of MMAE with IR increase immune cytotoxic cells. As corollary of these observation is that, using syngeneic (immunocompetent) models it's demonstrated that treatment with clinical approved check point inhibitor a-PD1 mAb increases MMAE radiosensitization. This part of the work is essential and novel and should be better defined.

Overall, this is an interesting work with relevant potential clinical implication, however additional data needed to support the raised conclusions before publication in Nature Comm.

My major concerns are the following:

- authors should provide direct evidence that MMAE represent an advantage respect to other approved payload, including MMAF, and discuss whether as example bystander effect or type of linker may have an impact on efficacy of trimodal chemio-radio-immunotherapy.
- authors claim that in syngeneic model MMAE lacked therapeutic efficacy irrespective of the coupled tumor-directed delivery vehicle. However, comparison here is misleading, as they compare MMAE activity coupled to an antibody vs a penetrating peptide. Instead, in order to fully address the concept that auristatine MMAE has limited antitumor activity in a syngeneic model a more appropriate control should be used, that is a MMAE coupled to a different mAb targeting other antigen (receptor). In addition, this would reinforce the evidence about the potential of ADC therapy in combination with IR in immunocompetent mouse system.
- authors state that "Utilizing ADCs at a lower radiosensitizing doses has the potential to achieve tumor kill while decreasing debilitating off-target systemic ADC toxicities." However, this notion cannot be extrapolated from data presented in this work. All the in vivo data were performed with for CDX3379-MMAE and ACPP-MMAE dosage ranging from 2.5 to 10 nM, which is not a suboptimal dosage. Moreover, it 's not exactly clear how in each single experiment dosage for the ADC or ACPP was chosen. More important, a dose escalation showing that ADC at a low dosage is able to increase IR activity is not present in this study.
- finally, to give more relevance to translational prospective of this study and strength the novel paradigm proposed by the authors, i.e. the efficacy of trimodal therapy, an additional HER3+ syngeneic model other than melanoma B16 should be used. Also it would be important to include the naked CDX mAb in the trimodal experimental arm in order to demonstrate the advantage of CDXADC vs naked CDX . Indeed, it cannot be excluded that naked CDX antibody it self has the ability to increase immune response in this context.

Specific comments:

Figure 1e: HER3 surface expression should be evaluated by FCAS.

Figure 2b: in this ADC drug release assay control tissue is missing and therefore, off target payload accumulation could not be excluded.

Figure 2c: these data seem to be out of the scope of this work.

Figure 2d-e: authors claim that antitumor efficacy of CDX3379-MMAE is modest in syngeneic model. However, there are no information about CDX3379 antibody affinity to mouse HER3. Could be explained this lack of efficacy with lower ability of CDX3379 antibody to accumulate in mouse tumors? In addition, authors do not comment at all that MMAF IC50 in mouse cell lines is significantly higher than human cell lines (as reported in Figure 1a).

Figure 3e: time line is missing in the treatment schema. No information are reported about cut-off value.

Figure 3g: the effect shown in this orthotopic model appear to be very modest. Statistical analysis is not clear and should be better defined. The $P < 0.01$ value should be specified whether is referred to differences between combo arm vs control .

Figure 4c: panel description in the figure should be improved. Moreover, statistic is missing in right panel.

Figure 4d-e-f: data are not clear as presented. For example: KM shown in (d) and tumor growth curve shown in (e) come from the same experiment or separate in vivo assays? Data in panel (f) could be eliminated as IgG control for antiCD8 is missing and similar data obtained with CDX3379-MMAE are definitively more convincing.

Figure 5b: growth curve showed in the top panel (Cytotoxic therapy) come from the same exp presented in Figure 3 e? KM curves for survival should be added. anti-PD1 dosage should be included in figure legend.

Figure 5c: data presented here for MC38 model are in disagreement with data shown in figure 3e. Indeed, combo treatment (ACPP-MMAE + IR) seems to have different activity in the two assays. Authors are invited to comment this discrepancy.

Figure 5b-5e-5f: in vivo assays shown in these panel come from the same B16 model. It would be helpful to combine all these data in one single panel showing antitumor efficacy in a more homogenous way, that is using KM survival curves.

Figure 5b-f: Authors should show data about the tolerability of the combination treatment.

Reviewer #2 (Remarks to the Author): with expertise in radio- and immune-therapy

In this manuscript, Hingorani et al., aim at investigating the efficacy of a trimodal precision chemo-radio-immunotherapy using spatially targeted auristatin warheads. To support this concept, authors have compiled a large body of evidence from cell biology, mass cytometry and in vivo experiments which suggest that the combination of auristatin antibody drug conjugates (ADC) with RT and immunotherapy spatially restrict the cytotoxicity of tumor kill of chemo-radiotherapy and then elicit anti-tumor immune responses.

Considering the widespread interest in combinatorial-based approaches to improve the fraction of patients that can benefit from immunotherapy (IT), this work is per se relevant and of major interest for the readers of Nature Communications. Overall, the manuscript is well-written but lack of detail and the inconsistencies in the radiation dose used is distracting from the main conclusion. Consequently, substantial reformatting of the manuscript together with additional experiments are warranted to ensure the adequate interpretation of their findings.

Comments:

- For all tumor growth curve and survival animal, indicate if the x axis correspond to day since the beginning of the treatment of the tumor cell implantation.

- Individual tumor growth curves that correspond to the survival data of B16 and MC38 mice experiment should be presented in the manuscript.
- The radiation method (instrument, dose, dose rate) is not described
- For all flow data, how the cells population were defined and the gating strategy should be shown.
- Fig 3e: the statistical analysis is misleading. Authors should compare ACP-MAE vs ACP-MAE+IR for all the tumor models.
- Fig 4d: the statistical analysis is misleading. Authors should compare IR vs ACP-MAE+IR
- No statistical analysis are presented for Fig 4c (CD8+ T cells)
- Fig 5c: the statistical analysis is misleading. Authors should compare ACP-MAE+IR vs ACP-MAE+IR+anti-PD1
- The fact that authors used different radiation regimens to combined with auristatin-ADC is very confusing. For instance, B16-tumor bearing mice received 5Gyx3 while LL/2 and MC38 tumors received 3Gyx3 (Fig 3e). No clear rationale is supporting the choice of this radiation regimen neither any indication on why RT was performed on days 3, 6 and 11 for B16 or on day 10 for MC38 (Fig 5a and 5c). This is especially important because different radiation regimens have been shown to elicit distinct immune stimulation.
- Along similar lines, the radiation regimens that are used are sub-optimal and could explain by the combination of IR+a-PD-1 is not efficacious (Fig 5f). Authors should increase the dose to 20Gy at least in the B16 model to recapitulate some of the data that are already available (Twyman-Saint Victor, Nature, 2015) and interrogate whether the addition of auristatin-ADC is augmenting this effect.
- It is also not clear how the sequencing between IR, auristatin-ADC and PD-1 blockade has been decided. For instance, for M38 tumors, anti-PD-1 is administered on day 10,13 and 18 while for B16, it is administered on day 3-6-11. Indeed, it has been demonstrated recently that blockade of PD-1 after completion of RT is leading to more efficacious (wei et al., Science Immunology). Thus, the lack of synergistic effect btw IT and anti-PD1 could be explained by the fact that observed in fig 5b can be explained by the fact that authors have blocked PD-1 during RT.
- Fig 5c: Not clear why the survival of MC38 is ending at day 30. Did the mice treated with the combination survived after that point? If so, did you rechallenge the animal? Same question applies to the B16 mice data of fig 5e.
- Supplementary Figure 8b. Remove CD8+, Tregs, NK, DC and M2 graphs because they are already shown in Figure 2b.

Reviewer #3 (Remarks to the Author): with expertise in cancer immunology/immunotherapy

This study reports a novel combination of radiosensitizing monomethyl auristatin E (MMAE) with tumor-directed antibodies. The study is very well planned and many syngeneic tumor models were tested, which substantiated the scientific rigor. The analysis of tumor immune infiltrate was thorough. The materials and methods included the experimental details, which further enhanced the rigor of the study. The manuscript is well written and easy to follow. There are several considerations that may be helpful to further improve the study.

1. MMAE is a radiosensitizing agent. MOC2 showed a modest response to CDX3379-MMAE in Figure 2d. Does the addition of IR in this model improve the response? This model was not tested in Figure 3.
2. The Y axes in Figure 2d are the absolute count numbers, which should be normalized to generate percentages as the total number of TILs could be very different among groups. The absolute counts do not reflect the population representation.
3. CDX3379-MMAE appears to significantly induce Tregs, which inhibit antitumor immune response. It would be helpful to examine the function of CD8 T cells in addition to the raw counts.
4. Figure 4a needs to show statistics of each population. The figure legends need to include details of gating.
5. Figure 4b shows both anti-tumor immune subsets and pro-tumor immune subsets (eg neutrophils) were increased by ACP-MAE-IR. More flow cytometric analysis on the functional status of CD8 T cells, CD4 T cells, and macrophages/DC would be helpful to strengthen the analysis.
6. In tumor models where ACP-MAE had a relatively modest effect (e.g. MOC2), does ACP-

MMAE induce a similar immune infiltration pattern as B16?

7. The figure legends should include how many times the in vivo experiments were repeated and the number of animals.

Overall, this is an exciting study with a comprehensive characterization of new technology.

Reviewer #4 (Remarks to the Author): with expertise in antibody drug conjugates, activatable cell penetrating peptides

In this manuscript, authors describe how different cancer models (xenografts and syngenic and autologous models) respond to targeting with locally activatable cell penetrating peptides and HER3 antibodies, conjugated to monomethyl auristatin. They show that combining targeted MMAE-delivery with radiotherapy and immune checkpoint inhibition, gives survival advantage in immune competent cancer models, while in xenografts in immune-compromised nude mice there is good activity of targeted MMAE alone.

General comments:

The manuscript is well written, but what is confusing is the large number of tumor models (human xenografts in mouse, mouse grafts in syngeneic models, autologous tumors) that is used in different settings, which makes appropriate comparisons difficult. Although I appreciate that confirmation in multiple models and systems is needed to make substantial conclusions, the study would have benefited from a comparison on the effects of HER3/ACPP-MMAE effects on B16 (or MOC2 or LL2) in immune-compromised and immune-competent mice: whereas CAL27 xenografts in nude mice respond well to MMAE targeting alone, with MMAE uptake in the tumors of roughly 100 nM (fig 2b), same intratumoral concentrations were seen in LL2 and B16 tumors in immune competent mice, but MMAE is grossly inactive in this setting.

Question that pops up is how sensitive LL2 and B16 cells are to ACPP-MMAE in vitro. Figure 1a shows that sensitivity of these cells to MMAE is much lower than of CAL27, but IC50s are still well below the 100 nM that is achieved in vivo. There is no good explanation for the fact that cells do not suffer from these high MMAE concentrations. Are these cells now stalled in G1? An important experiment would be to investigate how B16 or MOC2 cells respond to ACPP-MMAE when grown in nude mice, to elucidate the role of the immune system. Is ACPP-MMAE also affecting (phagocytic) immune cells, and possibly scavenged by these cells? Investigating proliferation/apoptosis indices in cancer cells and infiltrate could shed light on this.

Throughout the manuscript it is not clear what controls represent. E.g. in figure 1g, is control just non-injected, or sham injected? A proper control would be an MMAE-conjugate of a control antibody without targeting capacity. All too often signals like this occur because of increased vascular leakage in tumors. This phenomenon should be excluded. Also, the benefit of targeting with HER3 or ACPP should be demonstrated by comparison with free MMAE in comparable steady state levels.

With the main trimodal warhead approach, an important control is missing. Radiotherapy is increasingly used in combination with immune checkpoint inhibitors, with radiotherapy inducing formation of neo-antigens that evoke a T-cell infiltrate which can be activated by immune checkpoint inhibitors, as authors also refer to in the discussion. Without this important dual combination, a conclusion on the added value of targeted MMAE cannot be made.

Specific comments:

- In fig 1b, the macrophage marker CD68 goes up in B16 after 5 nM MMAE for 72 hr. When looking at the IC50 (1 nM or so?), I suppose that after 72 hr there will be significant cell kill. What was the quality of RNA? Then, CD68 is involved in lysosomal trafficking and autophagy, a pathway that is however negatively enriched in the pathways. Authors should discuss this discrepancy. Another issue is that authors state that after MMAE treatment the angiogenesis pathway is enriched. Authors should rephrase this: angiogenesis involves endothelial cells and perivascular cells, therefore such statements can only be done in a tumor context, not in isolated cell lines.

- CAL27 xenografts in immune compromised mice respond to anti-HER3 alone (Fig 2c) suggesting that the antibody also blocks intracellular HER3 signaling, independent from MMAE. Including MMAE now results in complete shutdown of growth of CAL27 tumors, but not of the CAL27 HER3 mutant

ones. How can MMAE-resistance of a subpopulation of the mutant cells be explained? MMAE effects are completely downstream of the HER3 pathway. How do CAL27wt and mutant cells respond to targeted MMAE in vitro?

Line 266: fig 3e: the **** significance; is this IR+MMAE vs IR, or vs MMAE? Please clarify

- Experiments with RAG mice showing that CD8+ T-cells are involved in the response to MMAE+IR are intriguing. A mechanistic explanation is however lacking. If T-cells are crucial for tumor control under IR, why is there such good response in CAL27 xenografts in nude mice with monotherapy only?

Line 122: Fig 1e relates to cell lines, not to xenografts as stated in the text

Line 183: please change title as here also immune-compromised models are tested.

In syngeneic MOC tumors there is no activity of CDX3379-MMAE at all. How do MOC cell lines in vitro respond to CDX3379 and CDX3379-MMAE? Is HER3 in MOC cells a driver, or are there alternative kinases that converge in the same signaling pathways to rescue the CDX3379 effects.

Reviewer #5 (Remarks to the Author): with expertise in antibody drug conjugates, cancer therapy

Key results and significance- The authors of the manuscript provide a detailed and comprehensive study that not only reports the synergistic effect of the MMAE-conjugates with IR for treating HER3 +ve tumours but have also detailed the mechanism that is necessary for immune-mediated responses, namely the role of the adaptive arm of immunity (through the need for functional CD8+ T cells) and the presence of checkpoint inhibition. The combination approach tested for tumour control was good evidence as a proof-of-concept study that advances the understanding of the principle behind how the tri-modal therapeutic strategy that was assessed works in vivo.

Data and Methodology- There were no notable concerns regarding the methodology that was followed and how they were described in the paper. The panel of results presented in the manuscript and the supplementary information was adequate. The interpretation of the results was also satisfactory and consistent. The data presented was thus convincing.

Analytical approach- The approach employed by the authors for the study was conceptually sound. The statistical tests employed were adequate and convincing.

Suggested improvements- The authors address the choice of antibody very well (lines 118-121) however it would also be relevant to briefly provide the reason behind the choice of the cell surface target HER3 over other well-characterized targets against which targeted antibodies are available as well, such as epidermal growth factor receptor (EGFR).

Clarity and context- The manuscript was clear, concise, and adequately reports findings with respect to work reported in the past with one minor exception: 132-133, "There are limited pre-clinical studies evaluating the efficacy of MMAE within the immune context"; providing the specifics of what is known about MMAE-mediated immune modulation and the exact knowledge gap that the authors aim to address through the study would improve the clarity of the section.

References- The authors of the manuscript have cited existing literature relevant to the study in a manner that was not only relevant but also comprehensive as well.

Response Letter to Reviewers

Dear Reviewers,

Thank you for your reviews of our manuscript. We thank you for the time taken and thoughtful, in-depth critiques of our initially submission. We have worked diligently to address the issues raised and have significantly revised our manuscript in response by providing additional data and improving clarity. We provide new data addressing the points of ADC specificity, ionizing radiation dosing and baseline tumor control of immune checkpoint inhibitors with ionizing radiation. In addition, Reviewer #1 raised the tantalizing idea of testing MMAE to MMAF auristatin warheads in the context of irradiate microenvironments, and we have now included this as new data which we believe to be the 1st time this has been done. Reviewer #2 brought up the idea of challenging mice cured by MMAE radiosensitization with tumor cell re-injection, and we have generated new data that further corroborates the role of adaptive immunity in mediating the effects of MMAE radiosensitization by demonstrating MMAE radiosensitized tumor kill results in immunologic memory. Below is a point by point response to the comments raised which are also marked in yellow highlight of the revised manuscript. Again, I truly appreciate the thoughtful and highly constructive feedback of your critiques of our initial submission which has helped craft a stronger manuscript.

Reviewer #1

Summary Comments: “In this work by Hingorani et al authors explore the potential of the trimodal precision chemo-radio-immunotherapy using a combination of MMAE loaded vehicles, IR and clinical approved anti-PD1 mAb. Although the finding of this work are significant, especially from a clinical translation perspective, how the entire manuscript has been set is not entirely convincing. In the first part of the paper, Figure 1,2 and 3 authors show data about activity of an anti-HER3 ADC or activable cell penetrating peptide (ACPP) coupled with MMAF alone and/or in combination with IR. These data are just a repetition of what the same group already presented in two recent papers: 1) 2016 (Adams et al, NatComm DOI: 10.1038/ncomms13019) where, using other ErbB receptor as targets (i.e. HER1 and HER2) it was consistently demonstrated that MMAE based ADCs are endowed with the ability to restrict payload radiosensitization to the tumor cells/tissue. 2) 2020 (Hingorani et al, Biomaterials <https://doi.org/10.1016/j.biomaterials.2020.120032>), where it's described the advantages of protease triggered cell penetrating peptide scaffolds to efficiently deliver radiosensitizing payloads (such as MMAE) increasing efficacy of radiotherapy. In summary, the data presented in the first part of this paper are not novel and possibly out of the scope of this paper. Indeed, the novelty here is represented by the mechanistic definition of how coupled MMAE is able to control tumor growth by radiosensitization of cancer cells. In this aspect authors reveal that payload radiosensitization and tumor control is mediated by adaptative immunity. In particular, they demonstrated that the combination of MMAE with IR increase immune cytotoxic cells. As corollary of these observation is that, using syngeneic (immunocompetent) models it's demonstrated that treatment with clinical approved check point inhibitor a-PD1 mAb increases MMAE radiosensitization. This part of the work is essential and novel and should be better defined. Overall, this is an interesting work with relevant potential clinical implication, however additional data needed to support the raised conclusions before publication in Nature Comm.”

Response: We thank the Reviewer for their supportive and constructive comments. We believe Figs. 1, 2 & 3 are essential in that they provide important context and results necessary to substantiate the novel findings of MMAE activity within the irradiated tumor immune microenvironment. We have worked attentively to address issues raised and provide new data to support of our findings more rigorously.

Comment 1: “My major concerns are the following: authors should provide direct evidence that MMAE represent an advantage respect to other approved payload, including MMAF, and discuss whether as example bystander effect or type of linker may have an impact on efficacy of trimodal chemo-radio-immunotherapy.”

Response: While we believe testing other ADC warheads/linkers to be a bit out of the scope of the current set of studies where we have focused on elucidating how MMAE radiosensitization performs in syngeneic tumors and engages anti-tumor immune responses, we too were intrigued by how the MMAF auristatin derivative would compare to MMAE in syngeneic tumor models grown in immune-competent mice. As the Reviewer noted, our lab has recently tested MMAE and MMAF warheads in human tumor xenograft models with MMAF warheads having decreased permeability as free drug translating into reduced bystander effects¹. We agree it would be quite interesting to see how the MMAF warhead with reduced free drug permeability/bystander effect

compares to MMAE in immune-competent cancer models. We have now tested the hypothesis raised by the Reviewer head on and present new data (Fig. 3h). First, we successfully synthesized a CDX3379-MMAF ADC (Supplementary Figs. 2c & 2d). Then, we tested CDX3379-MMAE + IR to CDX3379-MMAE IR directly in B16 murine tumors. Interestingly, CDX3379-MMAE + IR proved superior to CDX3379-MMAF + IR (Fig. 3h). Given these findings, we hypothesize that MMAE mediated bystander effects improve radiosensitized tumor control when compared to MMAF tumor cell radiosensitization and continue with MMAE as our lead radiosensitizing auristatin warhead.

Comment 2: “authors claim that in syngeneic model MMAE lacked therapeutic efficacy irrespective of the coupled tumor-directed delivery vehicle. However, comparison here is misleading, as they compare MMAE activity coupled to an antibody vs a penetrating peptide. Instead, in order to fully address the concept that auristatin MMAE has limited antitumor activity in a syngeneic model *a more appropriate control should be used, that is a MMAE coupled to a different mAb targeting other antigen (receptor)*. In addition, this would reinforce the evidence about the potential of ADC therapy in combination with IR in immunocompetent mouse system.”

Response: In principle, we agree with the sentiment raised by the Reviewer to have another ADC targeting a different cell surface receptor. However, antibodies that target murine cell surface receptors and syngeneic receptor driven murine models are limited. For these reasons, there are only a few studies with MMAE in immune competent tumor models and in this set of studies we have tried to offer a unique solution to this problem by simultaneously conjugating ADC warheads to antibodies and cell penetrating peptides to create robust pre-clinical research tools that can rigorously evaluate ADC warheads in routinely used pre-clinical syngeneic cancer models irrespective of antibody chosen for targeting. We were fortunate to have access to anti-HER3 antibody CDX3379 (generously provided by Celldex) that recognizes both human and murine isoforms of HER3. We completely agree that our tumor targeted cell penetrating peptide delivers MMAE to tumors by a different mechanism of action compared to antibodies. However, we would highlight that are 2 drug delivery platforms (ADC and activate cell penetrating peptide (ACPP)) phenocopy each other on key findings of our manuscript and allowed robust testing of MMAE radiosensitization in commonly used pre-clinical syngeneic murine cancer models including LL/2, B16 and MC38. CDX3379-MMAE+IR and ACPP-MMAE+IR showed similar tumor control (Figs. 3e & 3h), CD8 T cells infiltration & necessity for durable tumor control (Figs 4b, 4f & 4g). Finally anti-PD-1 immune checkpoint inhibition potentiated tumor control of both CDX3379-MMAE+IR and ACPP-MMAE+IR (Figs. 5e & 5f).

Comment 3: “authors state that “Utilizing ADCs at a lower radiosensitizing doses has the potential to achieve tumor kill while decreasing debilitating off-target systemic ADC toxicities.” However, this notion cannot be extrapolated from data presented in this work. All the in vivo data were performed with for *CDX3379-MMAE and ACPP-MMAE dosage ranging from 2.5 to 10 nM, which is not a suboptimal dosage*. Moreover, it’s *not exactly clear how in each single experiment dosage for the ADC or ACPP was chosen*. More important, a dose escalation showing that ADC at a low dosage is able to increase IR activity is not present in this study.

Response: As to the question of dosage, they were empirically determined for ACPP and ADC based on initial CDX3379-MMAE ADC studies. In pilot studies, we tested 0, 1 and 5 nmoles of CDX3379-MMAE and found 1 nmole to be ineffective but 5 nmoles had anti-tumor activity. Since radiotherapy is given in a fractionated manner (3 consecutive days in many of our studies), we split the dose of 5 nmoles to 2 injections given every other day (2.5 nmoles per day x 2 doses) and concurrent with ionizing radiation (3 consecutive daily fractions of IR, schema Fig. 3e). On average 4 MMAE molecules are attached per antibody in our CDX3379-MMAE ADC. In contrast, ACPP has only 1 MMAE molecule attached per cell penetrating peptide. Therefore to maintain molar MMAE drug dose equivalence between ACPP and ADC, we used 10 nmoles of ACPP-MMAE and 2.5 nmoles CDX3379-MMAE with both drug delivery vehicle delivering 10 nmoles MMAE per i.v. injection into the blood. We have now described this rationale in the **Results**. As to the issue of suboptimal doses and lower effective radiosensitizing doses, we made this statement based on a commonly used rationale for combinatorial therapies. We tempered this by stating there exists a “potential”. We would bring attention that the doses of MMAE monotherapy (CDX3379-MMAE or ACPP-MMAE) failed to result in significant tumor control. Moreover, the IR dose fractionation used as monotherapy also failed to produce durable tumor control in our cancer models and could be considered “suboptimal”. Importantly, when we combined ineffective ACPP-MMAE or CDX3379-MMAE doses together with an ineffective IR fractionation, the combinatorial therapy improved durable control in multiple tumor models (Figs. 3 & 5).

Comment 4: “finally, to give more relevance to translational prospective of this study and strength the novel paradigm proposed by the authors, i.e. the efficacy of trimodal therapy, an additional HER3+ syngeneic model other than melanoma B16 should be used. Also it would be important to include the naked CDX mAb in the trimodal experimental arm in order to demonstrate the advantage of CDXADC vs naked CDX. Indeed, it cannot be excluded that *naked CDX antibody itself has the ability to increase immune response in this context.*”

Response: To address naked antibody effects mechanistically, we tested if the CDX3379 ± IR influenced tumor immune infiltration and show this as new data (Fig. 4b). We focused on CD8 T cells since this immune cell population was most abundantly altered in MMAE radiosensitized tumors (Fig. 4a) and CD8 T cell depletion abrogated durable tumor control observed with MMAE radiosensitization (Figs. 4f & 4g). CDX3379 antibody alone did not alter CD8 tumor cell immune infiltration compared untreated mice and CDX3379 antibody + IR CD8 T cell infiltration was not significantly different than IR alone. Importantly, CDX3379 antibody + IR produced significantly less CD8 T cell infiltration compared to CDX3379-MMAE + IR. Also tackling this issue head on are new data comparing CDX3379-MMAE vs CDX3379-MMAF with IR (Fig. 3h). The only variable in these two treatments is the auristatin warhead (MMAE vs MMAF) conjugated to CDX3379 antibody. Both groups were treated with same dose of IR (5 Gy x 3) and CDX3379 antibody dose (2.5 nmoles x 2). Importantly, MMAE and not MMAF was necessary to improve tumor control giving credence to the critical contribution of MMAE for the radiosensitization seen with CDX3379-MMAE ADC. As to achieving more translational relevance, we believe that the larger translational prize is the universal findings of MMAE radiosensitization within the immune context occurs independent of drug delivery platform used to target MMAE. MMAE radiosensitization increases tumor control, stimulates tumor immune infiltration, achieves durable tumor control in a CD8 T cell dependent manner and is potentiated by anti-PD-1 immune checkpoint inhibition as either a CDX339-MMAE or ACPM-MMAE conjugate (Figs. 3e vs 3h, Figs., 4f vs 4g, and Figs. 5e vs 5f). Taken together, it suggests that our findings of MMAE radiosensitization within the tumor immune microenvironment may be more generalizable and broadly applicable to MMAE armed ADCs and not specific to targeting HER3. We believe this is the one of key strengths of our studies as we provide an alternative approach to characterize and validate ADC warheads in pre-clinical models that can then be studied in more depth with specific ADCs in advanced clinical development.

Specific Comment 1: “Figure 1e: HER3 surface expression should be evaluated by FCAS.”

Response: This is now shown in Fig. 2f and Supplementary Fig 3d.

Specific Comment 2: “Figure 2b: in this ADC drug release assay control tissue is missing and therefore, off target payload accumulation could not be excluded.”

Response: We wanted to confirm CDX3379-MMAE released drug within tissue. With regards to tumor specificity, we show 2 new pieces of data. First, we show Cy5 labeled CDX3379-MMAE accumulates in HER3 positive MOC2 tumors but not in HER3 negative LL/2 tumors (Fig. 2g, right panels). More importantly, we show target specificity of ADCs by using ADCs that do and do not recognize murine cell receptor targets (Figs. 2e & 2f). Murine HER3 targeted CDX3379-MMAE binds HER3 positive cells has activity against syngeneic tumors while cetux-MMAE ADC (human EGFR targeted cetuximab-MMAE conjugate) failed to bind to murine tumor cells and lacked activity compared to untreated control tumors.

Specific Comment 3: “Figure 2c: these data seem to be out of the scope of this work.”

Response: We believe these data validate that our synthesized CDX3379-MMAE ADC efficiently releases drug in tumors and is efficacious irrespective of HER3 downstream signaling. Given that this is the 1st description of CDX3379-MMAE, we believe showing its efficacy in human tumor xenografts has value to compare to the published literature with MMAE ADCs that are exclusively tested in human xenografts. The CAL27 PIK3CA further substantiates the efficacy of CDX3379-MMAE to release drug and kill cells independent of HER3 signal inhibition by the antibody.

Specific Comment 4: “Figure 2d-e: authors claim that antitumor efficacy of CDX3379-MMAE is modest in syngeneic model. However, there are no information about CDX3379 antibody affinity to mouse HER3. Could be explained this lack of efficacy with lower ability of CDX3379 antibody to accumulate in mouse tumors?”

Response: The therapeutic efficacy in murine syngeneic tumor models with CDX3379 antibody alone has been published². In our own studies which used reduced doses of CDX3379, Cy5 labeled CDX3379-MMAE binds to both human and murine tumors in situ (Figs. 1f & 1g). In using the term “modest”, we were merely trying to convey the fact that while CDX3379-MMAE efficiently controlled human xenograft tumors while a similar dose had limited activity in syngeneic tumors (Figs. 2c-e). In addition, the reduced activity of MMAE in

murine syngeneic tumors does not appear to be an issue of HER3 affinity of CDX3379 since MMAE delivered to tumors as a cell penetrating peptide conjugate (i.e. ACP-PP-MMAE) also failed to increase tumor control in multiple cancer models (Figs. 2g & 3e).

Specific Comment 5: “In addition, authors do not comment at all that MMAF IC₅₀ in mouse cell lines is significantly higher than human cell lines (as reported in Figure 1a).”

Response: The Reviewer astutely notes that the IC₅₀ of MMAE is higher in murine cancer cells compared to human cancer cells. In Fig. 1a, our objective was to ensure that murine cancer cells were sensitive to MMAE exposure as we transitioned studies from human tumor xenografts to syngeneic murine tumors to understand how MMAE radiosensitization engaged anti-tumor immune responses. While MMAE is less potent in murine cancer cells compared to human cells, MMAE remains more potent than conventional chemotherapies in murine cells. We have made note in the Results that MMAE showed decreased potency in murine cancer cells compared to human cancer cells.

Specific Comment 6: “Figure 3e: time line is missing in the treatment schema. No information are reported about cut-off value.”

Response: The timeline schema in Fig 3e is meant to orient the reader to sequencing of MMAE and IR. On day of therapy initiation, “Carrier-MMAE” (i.e. ADC or ACP-PP conjugate) is injected. IR is started on the following day and given for 3 consecutive days. A 2nd dose of conjugated MMAE is given one day after the 1st dose of MMAE is injected. This treatment delivery schema was used in all studies with MMAE and IR in subcutaneous syngeneic tumors (i.e. LL/2, B16 and MC38). If “cut-off value” refers to experiment terminating at day 60, our initial pilot studies went as far as day 90 from day of tumor cell implantation. However, mouse husbandry costs and vivarium space constraints limited length of therapy studies. In studies where long term tumor control was seen, we reported studies at day 30 or 60 depending on hypothesis being tested in experiments.

Specific Comment 7: “Figure 3g: the effect shown in this orthotopic model appear to be very modest. Statistical analysis is not clear and should be better defined. The P<0.01 value should be specified whether is referred to differences between combo arm vs control.”

Response: We agree that the effect appears limited in this autochthonous carcinogen induced cancer model. However, we do believe there is value in presenting this data as it further validates the efficacy of MMAE radiosensitization in a unique cancer model that more closely mimics how human cancers more slowly evolve in patients. P value comparison is between combo arm and control and now specified.

Specific Comment 8: “Figure 4c: panel description in the figure should be improved. Moreover, statistic is missing in right panel.”

Response: We apologize for this omission and statistical analysis is now shown (also brought up by Reviewer #2). In the revised manuscript, this data is now shown in Supplementary Fig. 10c with updated Figure Legend to better describe studies.

Specific Comment 9: “Figure 4d-e-f: data are not clear as presented. For example: KM shown in (d) and tumor growth curve shown in (e) come from the same experiment or separate in vivo assays? Data in panel (f) could be eliminated as IgG control for antiCD8 is missing and similar data obtained with CDX3379-MMAE are definitively more convincing.”

Response: We have significantly reworked Fig. 4 based on Reviewers’ suggestions to improve clarity and new data generated to better support our conclusions. The key conclusion we wished to convey in initially submitted Figs. 4d & 4e (resubmission Supplementary Fig. 10d and Main Fig. 4e) is that tumor control of ACP-PP-MMAE + IR seen in fully immune-competent WT C57BL/6 mice (Fig. 3) is significantly curtailed when tumors are grown in immune-deficient RAG1 KO mice but still treated with ACP-PP-MMAE and IR. The Reviewer is correct that data in Fig. 4e came from 4d. To improve clarity and focus on key findings, we moved the RAG1 KO survival curve of the 4 treatment groups (old Fig. 4d) to Supplementary Fig. 10d. Now, we only show a direct head to head comparison of individual tumor growth response to ACP-PP-MMAE + IR in immune-competent C57BL/6 WT mice vs immune-deficient RAG1 KO mice lacking mature B and T cells (Fig. 4e) driving home the point that adaptive immunity plays a role of how tumors respond to MMAE radiosensitization. Following these results in Fig. 4e, we then show the necessity of CD8 T cells for sustained tumor control by MMAE + IR (Figs. 4f & 4g). We believe showing necessity of CD8 T cells for prolonged tumor control in mice treated with MMAE + IR occurs independent of drug delivery platform, i.e. ADC or ACP-PP conjugate, side by side helps establish scientific rigor and the generalizability of our results in that it is not

limited to our HER3+ targeted MMAE ADC (CDX3379-MMAE), but rather an inherent property of MMAE in the irradiated tumor environment (Figs. 4f & 4g).

Specific Comment 10: “Figure 5b: growth curve showed in the top panel (Cytotoxic therapy) come from the same exp presented in Figure 3 e? KM curves for survival should be added. anti-PD1 dosage should be included in figure legend.”

Response: We have done our best to ensure our results are reproducible and have replicated key findings presented in independent experiments using multiple tumor models and two different MMAE drug carriers (i.e. cell penetrating peptides and ADC). All data in our new Fig. 5b is not data in Fig. 3e but instead comes from independent studies. Given that 8 treatment groups need to be presented to fully evaluate all treatment permutations of MMAE ± IR ± anti-PD-1 antibody, we have presented the data in the 3 models differently to give the reader different vantage points. Also based on tumor sensitivities in the different models and time frames of studies, we have tried to present the data in a manageable manner for both the in depth reading audience and those who prefer a quicker overview. In the B16 model, we show individual tumor responses of mice (Fig. 5b). In line with the Reviewer’s suggestion, we show survival curves of the key treatment groups (ACPP-MMAE + IR +/- anti-PD-1) in Fig. 5e of data presented in Fig. 5b (see response below to Reviewer #1, Specific Comment #12). We have noted in **Figure Legend** that the data used to generate survival curves in Fig. 5e comes from data in Fig. 5b. In the MC38 model, we also present data as suggested by the Reviewer as survival curves (Fig. 5c). Finally in the orthotopic 4MOSC1 tongue model, we show data on last day of study of when majority of mice needed to be sacrificed due to tumor burden (Fig. 5d). We have added anti-PD-1 dosage in Figure Legend to Fig. 5 as requested.

Specific Comment 11: “Figure 5c: data presented here for MC38 model are in disagreement with data shown in figure 3e. Indeed, combo treatment (ACPP-MMAE + IR) seems to have different activity in the two assays. Authors are invited to comment this discrepancy.”

Response: As the Reviewer astutely points out, MC38 tumor responses to cytotoxic therapy in absence of anti-PD-1 therapy differs in Figs 3e & 5c. MC38 tumors showed robust response to ACPP-MMAE + IR (Fig. 3e) which allowed us to test this model in a tumor re-challenge experiment (Fig. 4h) since a majority of mice had complete tumor regression following ACPP-MMAE + IR. In the MC38 tumor therapy studies in Figs. 3e & 4h, treatment was initiated on day 6 after tumor cell implantation and majority of tumor were “cured”. To be able to validate if anti-PD-1 could potentiate MMAE + IR in MC38 tumors, we delayed start of tumor therapy to day 10 post tumor cell implantation, which allowed 4 days additional of tumor growth and made it harder to control tumors with ACPP-MMAE + IR. Starting therapy with such larger tumors allowed us to show anti-PD-1 potentiation in MC38 tumors and validated the advantage of trimodal therapy as was also seen with 2 additional cancer models, B16 and 4MOSC1.

Specific Comment 12: “Figure 5b-5e-5f: in vivo assays shown in these panel come from the same B16 model. It would be helpful to combine all these data in one single panel showing antitumor efficacy in a more homogenous way, that is using KM survival curves.”

Response: The data in Figs. 5b & 5e come from the same experiment. We analyzed the individual tumor volume data of B16 response to ACPP-MMAE + IR vs ACPP-MMAE + IR + anti-PD-1 (Fig. 5b) as survival curves (Fig. 5e) so that it could be evaluated side by side with an independent experiment testing CDX3379-MMAE + IR vs CDX3379-MMAE + IR + anti-PD-1 (Fig. 5f). We believe there is value in showing Figs. 5e & 5f next to each other of anti-PD-1 potentiating MMAE radiosensitization independent of how MMAE is delivered to tumors. As with MMAE therapy radiosensitization (Figs. 3e & 3h) and CD8 depletion studies (Figs. 4f & 4g), we are able to phenocopy and validate the efficacy of combining MMAE radiosensitization with anti-PD-1 antibody irrespective of MMAE drug deliver platform, i.e. ACPP conjugate or ADC (Figs. 5e & 5f) which provides support for the broader applicability of MMAE radiosensitization engaging anti-tumor immune responses. Our results are not limited to CDX3379-MMAE and HER3+ tumors but rather should be translatable to the broader pipeline of MMAE ADCs in various stages of clinical development.

Specific Comment 13: “Figure 5b-f: Authors should show data about the tolerability of the combination treatment.”

Response: We did not notice any adverse effects or unexpected toxicity of MMAE radiosensitization ± anti-PD-1 antibody. We have included representative mouse body weights of studies as Supplementary Figs. 9d, 9e & 9f.

Reviewer #2

Summary Comments: “In this manuscript, Hingorani et al., aim at investigating the efficacy of a trimodal precision chemo-radio-immunotherapy using spatially targeted auristatin warheads. To support this concept, authors have compiled a large body of evidence from cell biology, mass cytometry and in vivo experiments which suggest that the combination of auristatin antibody drug conjugates (ADC) with RT and immunotherapy spatially restrict the cytotoxicity of tumor kill of chemo-radiotherapy and then elicit anti-tumor immune responses. Considering the widespread interest in combinatorial-based approaches to improve the fraction of patients that can benefit from immunotherapy (IT), this work is per se relevant and of major interest for the readers of Nature Communications. Overall, the manuscript is well-written but lack of detail and the inconsistencies in the radiation dose used is distracting from the main conclusion. Consequently, substantial reformatting of the manuscript together with additional experiments are warranted to ensure the adequate interpretation of their findings.”

Response: We thank the Reviewer for their support and constructive feedback. With regards to ionizing radiation doses used, we have standardized radiation doses as best we could and have provide rationales for alterations in dosing due to radiosensitivity, hypothesis tested and/or logistics. In particular, we have new data with LL/2 tumor control by MMAE and IR using 5 Gy fractions (old Fig. 3e used 3 Gy fractions) and plotted the data as a survival curve to be consistent with B16 and MC38 survival plots and allow for easier comparison of the three syngeneic tumor models in new Fig. 3e.

Comment 1: “For all tumor growth curve and survival animal, indicate if the x axis correspond to day since the beginning of the treatment of the tumor cell implantation.”

Response: We have now clarified this in the Figure Legends. Day 0 on tumor survival and growth curve X-axis refers to time of tumor cell implantation. The only exception is the MOC1 tumor survival curves in Fig 2e, where tumors required longer to establish prior to treatment initiation.

Comment 2: “Individual tumor growth curves that correspond to the survival data of B16 and MC38 mice experiment should be presented in the manuscript.”

Response: We have now included growth curves of the LL/2, B16 and MC38 tumors of data shown as survival curves in Fig. 3e. Given the large amount of data and number of groups across multiple tumor models, we have presented the data as mean tumor volumes of all surviving mice on days measured. This data is now shown in Supplementary Fig. 9d, upper panels.

Comment 3: “The radiation method (instrument, dose, dose rate) is not described.”

Response: We apologize for this omission and ionizing radiation delivery is now included in Methods.

Comment 4: “For all flow data, how the cells population were defined and the gating strategy should be shown.”

Response: Gating strategy with definition of cell populations is now included as Supplementary Fig. 10a.

Comment 5: “Fig 3e: the statistical analysis is misleading. Authors should compare ACP-MAE vs ACP-MAE+IR for all the tumor models.”

Response: The statistical analysis in our original submission compared all 4 experimental groups. Based on Reviewer’s suggestion, we have now updated statistical analysis in Fig. 3e. We apologize if the original analysis was misleading. For scientific rigor, we would like to highlight that tumor response studies in syngeneic mice were replicated multiple times in different tumor models and using different drug carriers i.e. ADC or ACP conjugate (Figs. 2, 3, 4 & 5) and consistently demonstrate the superiority of MMAE radiosensitization.

Comment 6: “Fig 4d: the statistical analysis is misleading. Authors should compare IR vs ACP-MAE+IR”

Response: Again, the statistical analysis in our original submission compared all 4 experimental groups. However based on the Reviewer’s suggestion we have now updated statistical analysis. Also based on Reviewer # 1’s comment (above Specific Comment #9), this figure (initial Fig. 4d) has been moved to Supplementary Fig. 10d.

Comment 7: “No statistical analysis are presented for Fig 4c (CD8+ T cells)”

Response: I apologize for this omission and statistical analysis is now shown. In the revised manuscript, this figure has been moved to Supplementary Fig. 10c.

Comment 8: “Fig 5c: the statistical analysis is misleading. Authors should compare ACP-MAE+IR vs ACP-MAE+IR+anti-PD1”

Response: Once again the initial statistical analysis shown involved 4 treatment groups. Based on Reviewer's suggestion we have now updated statistical analysis shown in Fig. 5c. We have also updated statistical analysis in Figs. 5d, 5e & 5f to consistently analyze MMAE + IR to MMAE + IR + anti-PD-1.

Comment 9: "The fact that authors used different radiation regimens to combined with auristatin-ADC is very confusing. For instance, B16-tumor bearing mice received 5Gyx3 while LL/2 and MC38 tumors received 3Gyx3 (Fig 3e). No clear rationale is supporting the choice of this radiation regimen neither any indication on why RT was performed on days 3, 6 and 11 for B16 or on day 10 for MC38 (Fig 5a and 5c). This is especially important because different radiation regimens have been shown to elicit distinct immune stimulation."

Response: We apologize for not clearly stating radiation regimens used. We have created more uniformity of ionizing radiation (IR) doses in this resubmission. For all syngeneic models from established tumor cell lines (LL/2, B16, MC38) grown subcutaneously, we used 3 fractions of IR given over 3 consecutive days to mimic clinical radiotherapy. The 3 daily fraction of IR were chosen based on the mechanism of action of anti-tubulin MMAE and our prior work on characterizing its radiosensitization³. Based on our prior work with MMAE and IR in human tumor xenografts, the 1st day of IR was the day after MMAE injection to allow for cells to accumulate in the radiosensitive G₂/M phase of the cell cycle^{3,4}. LL/2 and B16 tumors were treated 5 Gy fractions (15 Gy total). When we used the same 5 Gy per fraction dose in MC38 tumors, MC38 tumors exhibited increased radiosensitivity to IR alone with 5 Gy x 3 resulting in tumor regression. Therefore, we reduced the IR dose to 3 Gy per fraction from 5 Gy in the MC38 model (9 Gy total as opposed to 15 Gy for LL/2 and B16 tumors) so that we could test for MMAE radiosensitization. For orthotopic tongue 4MOSC1 and 4NQO tumors we limited irradiation to 1 or 2 fractions respectively to corroborate radiosensitization results and minimize mouse morbidity from oral cavity irradiation. We apologize for any confusion, RT was not given on days 3, 6 and 11 in Fig. 5b, instead those were days that anti-PD-1 antibody was given. We have further clarified these issues in the **Results, Figure Legends, and Methods**.

Comment 10: "Along similar lines, the radiation regimens that are used are sub-optimal and could explain by the combination of IR+a-PD-1 is not efficacious (Fig 5f). Authors should increase the dose to 20Gy at least in the B16 model to recapitulate some of the data that are already available (Twyman-Saint Victor, Nature, 2015) and interrogate whether the addition of auristatin-ADC is augmenting this effect."

Response: As the Reviewer astutely points out, larger ablative IR doses have been tested in published studies with immune checkpoint inhibition^{5,6}. As noted in the **Discussion**, we tested an approach that was conceptually different than such ablative strategies in that we attempted to leverage the highly successful concurrent chemo-radiotherapy paradigm by integrating immunotherapies with tumor targeted radiosensitization. The fractionated IR dose regimens (i.e. 3-5 Gy x 3) we employed produced modest growth delay in irradiated tumors as monotherapy with or without anti-PD-1 antibody. Importantly, MMAE combined with these smaller fractionated IR doses results in CD8 T cell dependent durable tumor control, induces immunologic memory and potentiates anti-PD-1 immune checkpoint inhibition. From a clinical translational standpoint, our approach offers an alternative paradigm for integrating immunotherapies with radiotherapy. Not all irradiated tumor sites can safely tolerate higher fraction size ablative SRS/SBRT IR doses, and we believe our approach provides a clinically relevant alternative approach to integrating immunotherapies with ADC based precision radiosensitization. That said, we agree that it will be worthwhile in future studies to directly compare/contrast immunotherapies with ablative IR doses vs targeted radiosensitized lower IR doses but believe them to beyond the scope of the current studies.

Comment 11: "It is also not clear how the sequencing between IR, auristatin-ADC and PD-1 blockade has been decided. For instance, for M38 tumors, anti-PD-1 is administered on day 10,13 and 18 while for B16, it is administered on day 3-6-11. Indeed, it has been demonstrated recently that blockade of PD-1 after completion of RT is leading to more efficacious (wei et al., Science Immunology). Thus, the lack of synergistic effect btw IR and anti-PD1 could be explained by the fact that observed in fig 5b can be explained by the fact that authors have blocked PD-1 during RT."

Response: We based our studies with immune checkpoint inhibition with MMAE radiosensitization based on published studies and our co-authors' experiences (MM Allevato & JS Gutkind)^{2,5}. Twyman-Saint Victor (Nature, 2015) showed concurrent checkpoint inhibition with IR produced anti-tumor efficacy in B16 melanoma models given as 3 doses and provided the rationale for our initial timing of anti-PD-1 antibody with MMAE and IR⁵. As to the apparent discrepancies of anti-PD-1 therapy delivery between B16 and MC38 tumors, anti-PD-1 therapy dosing was standardized to the of 1st day of MMAE injection which was day 3 in B16 tumors and day 10 in MC38 tumors. After the 1st dose of anti-PD-1, the 2nd dose and 3rd doses were given at equivalent time intervals in all models, 3 days, and 8 days after the 1st anti-PD-1 injection. We agree

that it will be worthwhile to test if temporal sequencing of immune checkpoint inhibition influence MMAE radiosensitized tumor control in future studies to optimize immune responses and tumor control.

Comment 12: “Fig 5c: Not clear why the survival of MC38 is ending at day 30. Did the mice treated with the combination survived after that point? If so, did you rechallenge the animal? Same question applies to the B16 mice data of fig 5e.”

Response: Based on our initial studies of tumor control being evident by day 30, vivarium space issues, and goals of studies, syngeneic tumor therapy studies were routinely ended on day 30 or day 60. The Reviewer does bring up an intriguing hypothesis, the rechallenge of “cured” mice to test for immunologic memory. We specifically tested this hypothesis and show this new data in Fig. 4h. Based on results shown in Fig 5e with MC38 tumor bearing mice, mice with MC38 tumor cells injected on day 0 were initially treated with ACP-MAE + IR as in Fig. 5e on day 6. Mice “cured” of tumors bilaterally were then challenged with re-injection of MC38 tumor cells on day 59. To control for tumor cell re-injection on day 59, a set of tumor naïve mice were also injected to ensure tumor cells formed tumors in mice. Excitingly, ACP-MAE + IR “cured” mice did not show tumor growth when challenged with re-implantation of tumor cells suggesting ACP-MAE radiosensitized tumor kill induces immunologic memory. We thank the Reviewer for suggesting this experiment as it strengthens our overarching premise of MMAE radiosensitization stimulating anti-tumor immune effects.

Comment 13: “Supplementary Figure 8b. Remove CD8+, Tregs, NK, DC and M2 graphs because they are already shown in Figure 2b.”

Response: My apologies, I thought it may be of use to have the complete data set in one figure for the reader to evaluate more easily. We have removed graphs shown in Fig. 2d from Supplementary Fig. 8b.

Reviewer #3

Summary Comments: “This study reports a novel combination of radiosensitizing monomethyl auristatin E (MMAE) with tumor-directed antibodies. The study is very well planned and many syngeneic tumor models were tested, which substantiated the scientific rigor. The analysis of tumor immune infiltrate was thorough. The materials and methods included the experimental details, which further enhanced the rigor of the study. The manuscript is well written and easy to follow. There are several considerations that may be helpful to further improve the study”

Response: We thank the Reviewer for their supportive comments of our studies.

Comment 1. “MMAE is a radiosensitizing agent. MOC2 showed a modest response to CDX3379-MMAE in Figure 2d. Does the addition of IR in this model improve the response? This model was not tested in Figure 3.”

Response: The MOC2 studies done in Fig. 2d using CDX3379-MMAE were done in orthotopic grown tongue tumors. We did perform a pilot study of adding IR to CDX3379-MMAE to the MOC2 orthotopic tumor model. However, all mice had to be quickly sacrificed due to tongue tumor burden causing distress and not allowing for long term evaluation. Moving to subcutaneous tumors allowed for longer timeframe evaluation of the MMAE radiosensitized tumor immune microenvironment where we tested LL/2, B16 and MC38 tumors with MMAE (ACP conjugate or ADC) and IR. All 3 models show an advantage to combining MMAE with IR, albeit with varying degrees of durable tumor control.

Comment 2: “The Y axes in Figure 2d are the absolute count numbers, which should be normalized to generate percentages as the total number of TILs could be very different among groups. The absolute counts do not reflect the population representation.”

Response: We agree with the Reviewer and the data plotted on the Y axis for each treatment condition was not the absolute cell number in total tumor samples but rather number of cells in equivalent events counted/sample. I apologize for this confusion. The data was normalized using Matlab-based normalization software based on the EQ bead-removal. Prior to analysis, samples were subsampled to the lowest number of events so all counts were normalized to a total number of events, i.e. 2324. We have presented normalized data as suggested by the Reviewer and clarified this in the **Figures** (Fig. 2d & Supplementary Fig. 8b) and **Methods**.

Comment 3: “CDX3379-MMAE appears to significantly induce Tregs, which inhibit antitumor immune response. It would be helpful to examine the function of CD8 T cells in addition to the raw counts.”

Response: The interplay of immune cells is an intriguing question. To begin to tackle this issue, we focused on CD8 T cells in more depth given the relative and absolute changes induced by MMAE radiosensitization. We have evaluated CD8 T cell expression levels of granzyme B, PD-1 and PD-L1 levels (Supplementary Fig. 10c & Fig. 5a). For CD8 T cells granzyme B expression, MMAE radiosensitization increased granzyme B at both

the RNA and protein levels in tumors. Investigating T cell exhaustion markers, IR alone appeared to be the key driver for CD8 T cell PD-1 and PD-L1 upregulation that was not further altered by MMAE radiosensitization. In future studies, we hope to dive deeper into how MMAE radiosensitization affects functions of CD8 T cells and other immune mediators.

Comment 4: “Figure 4a needs to show statistics of each population. The figure legends need to include details of gating.”

Response: I apologize for this omission. Details of our flow cytometry gating strategy are now shown Supplementary Fig. 10a. Statistics for each gated cell population is now fully shown in Supplementary Fig. 10b.

Comment 5. “Figure 4b shows both anti-tumor immune subsets and pro-tumor immune subsets (e.g. neutrophils) were increased by ACP-MAE-IR. More flow cytometric analysis on the functional status of CD8 T cells, CD4 T cells, and macrophages/DC would be helpful to strengthen the analysis.”

Response: CD8 T cells showed the most robust quantitative changes with MMAE+IR which was seen when MMAE was delivered either as an ADC and or ACP conjugate (Figs. 4a, 4b & 4d). To strengthen our findings, we have now more thoroughly interrogated CD8 T cell immune infiltration with multiple control conditions including CDX3379 antibody + IR (new Fig. 4b). In addition we have evaluated granzyme B expression of CD8 T cells and shown it to be elevated at both the RNA and protein levels (Supplementary Fig. 10c). It will be curious how MMAE radiosensitization functionally alters other immune cell types, but for now we believe they are beyond the scope of our present work.

Comment 6. “In tumor models where ACP-MAE had a relatively modest effect (e.g. MOC2), does ACP-MAE induce a similar immune infiltration pattern as B16?”

Response: Our immunologically colder tumor models (i.e. MOC2 and LL/2), showed modest anti-tumor activity with MMAE alone (delivered as an ADC for MOC2 and ACP conjugate for LL/2). Given the lack of anti-tumor effect, we have deferred characterizing immune infiltration in these models. Such immunologically colder tumors may need an alternative strategy to optimize engagement of anti-tumor immune responses following MMAE radiosensitization.

Comment 7: “The figure legends should include how many times the in vivo experiments were repeated and the number of animals”.

Response: The number of animals in each experiment are now included in **Figure Legends**. As the Reviewer noted, “many syngeneic tumor models were tested, which substantiates the scientific rigor. MMAE radiosensitization studies with B16, LL/2 and MC38 were done at least 3 times, but they were refined (radiation dose or MMAE dose) based on what we learned from the preceding studies to optimize integration of MMAE radiosensitization with immune checkpoint inhibition. While such studies are not true replicates, we believe they add to the scientific rigor of our overall findings.

Reviewer #4

Summary Comment 1: “In this manuscript, authors describe how different cancer models (xenografts and syngeneic and autologous models) respond to targeting with locally activatable cell penetrating peptides and HER3 antibodies, conjugated to monomethyl auristatin. They show that combining targeted MMAE-delivery with radiotherapy and immune checkpoint inhibition, gives survival advantage in immune competent cancer models, while in xenografts in immune-compromised nude mice there is good activity of targeted MMAE alone. General comments:

The manuscript is well written, but what is confusing is the large number of tumor models (human xenografts in mouse, mouse grafts in syngeneic models, autologous tumors) that is used in different settings, which makes appropriate comparisons difficult. Although I appreciate that confirmation in multiple models and systems is needed to make substantial conclusions, the study would have benefited from a comparison on the effects of HER3/ACP-MAE effects on B16 (or MOC2 or LL2) in immune-compromised and immune-competent mice whereas CAL27 xenografts in nude mice respond well to MMAE targeting alone, with MMAE uptake in the tumors of roughly 100 nM (fig 2b), same intratumoral concentrations were seen in LL2 and B16 tumors in immune competent mice, but MMAE is grossly inactive in this setting. Question that pops up is how sensitive LL2 and B16 cells are to ACP-MAE in vitro. Figure 1a shows that sensitivity of these cells to MMAE is much lower than of CAL27, but IC50s are still well below the 100 nM that is achieved in vivo. There is no good explanation for the fact that cells do not suffer from these high MMAE concentrations. Are these cells now stalled in G1? An important experiment would be to investigate how B16 or MOC2 cells respond to ACP-MAE when grown in nude mice, to elucidate the role of the immune system. Is ACP-MAE also affecting

(phagocytic) immune cells, and possibly scavenged by these cells? Investigating proliferation/apoptosis indices in cancer cells and infiltrate could shed light on this”

Response: As the Reviewer noted, a strength of our studies was the use of multiple models to comprehensively evaluate MMAE radiosensitization within the tumor immune microenvironment and demonstrate the range of effectiveness. We aimed to test our approach as broadly as possible across different cancer models for scientific rigor. We used isogenic human CAL27 xenografts to validate our HER3 targeted CDX3379-MMAE ADC kills tumor cells irrespective of sensitivity to HER3 signal inhibition. Since this is the 1st description of CDX3379-MMAE ADC, we wanted to provide strong evidence that it could successfully delivery active MMAE to receptor positive tumors irrespective of receptor signaling resistance. Moreover, our results in CAL27 human xenografts are in line with the wealth of published literature that ADCs are highly effective against receptor positive human xenografts. With regards to reduced efficacy of MMAE in syngeneic cancer models compared with human xenografts, we too were a bit surprised that MMAE lacked significant anti-tumor efficacy as a monotherapy across multiple syngeneic tumors that was corroborated irrespective of tumor targeted delivery vehicle (ADC, or ACP conjugate). Our results showing modest efficacy of MMAE monotherapy in syngeneic tumors is in line with a recent publication testing AXL targeted enapotamab vedotin⁷. The Reviewer raises a key question that it would be of value to evaluate MMAE ± IR in one tumor model grown in immune-compromised versus immune-competent mice. We completely agree and this served as motivation for our RAG1 KO studies and CD8 T cell depletion studies (Figs. 4e, 4f & 4g). To first test this, we established syngeneic B16 tumors in WT C57BL/6 mice and in RAG 1 knockout mice (Fig 4e and Supplementary Fig 10d). RAG1 knockout mice lack mature T or B cells and so we used this mouse model instead of nude mice lacking only T cells. Having established that T and/or B cells are necessary for durable tumor control, we more specifically show CD8 T cells are required for MMAE radiosensitized tumor control irrespective of how MMAE is delivered to B16 tumors (i.e. ADC or ACP conjugate) (Figs. 4f & 4g). As to the varied sensitivity of syngeneic tumors to ACP-MMAE ± IR (Fig. 5e). We tested the hypothesis that ACP-MMAE delivered MMAE less efficiently to LL/2. As the Reviewer noted, ACP delivered its MMAE cargo to LL/2 tumor with similar efficiency as B16 tumors (Fig. 3f). Our working hypothesis is that immunologically “colder” tumors such as LL/2 & MOC2 also have blunted anti-tumor immune responses to MMAE radiosensitization and in part explains reduced tumor control. This hypothesis is supported by data demonstrating that the efficacy of B16 radiosensitization is curtailed in RAG1 knockout mice (Fig. 4e) lacking mature B and T cells or specific depletion of CD8 T cells from immune-competent mice (Figs. 4f & 4g). As to the issue of testing ACP-MMAE in cell culture, we have previously shown that the ACP scaffold blocks MMAE uptake in cell culture while the r9 cell penetrating peptide within the ACP scaffold is taken up by LL/2, B16 and MC38 cells (Figure 2 from Hingorani et al *Pharmaceutics*, 2021, Figure attached to right⁸). And so we believe that the curtailed efficacy of ACP-MMAE + IR in LL/2 tumors compared to B16 and MC38 tumors is not due lack of ACP delivery to LL/2 tumor cells.

Figure 2. Activatable cell penetrating peptide (ACPP) scaffold blocks cell penetrating peptide (CPP) uptake by murine tumor cells. (A) Structural representation of Cy5 labeled CPP and ACPP. (B) Murine cancer cells exposed to Cy5 labeled CPP and ACPP probes. Cells imaged for Cy5 fluorescence (Magenta). Nuclei stained with DAPI (Blue).

Summary Comment 2: “Throughout the manuscript it is not clear what controls represent. E.g. in figure 1g, is control just non-injected, or sham injected? A proper control would be an MMAE-conjugate of a control antibody without targeting capacity. All too often signals like this occur because of increased vascular leakage in tumors. This phenomenon should be excluded.”

Response: I apologize for not clearly defining what “Control” meant in many of the figure panels. We have now corrected this throughout the manuscript. The Reviewer also brings up an important issue of ADC targeting specificity. We have now included new data to directly address this. First, we tested HER3 targeted CDX3379-MMAE tumor accumulation in mice bearing HER3 negative LL/2 tumors or HER3 positive tumors (MOC1, MOC2) (Fig 1e, 1g). These results demonstrate target specific tumor accumulation of CDX3379-MMAE in HER3+ MOC1 & MOC2 tumors but not in HER3 negative LL/2 tumors. To further validate CDX3379-MMAE target specificity and conjugate stability, we synthesized cetuximab-MMAE to serve as negative control for HER3+ syngeneic murine tumors. Cetuximab (Erbix) is a clinical grade antibody to human isoform of EGFR. While HER3 targeted CDX3379-MMAE attached to cell surface of murine B16 cells in dose dependent manner, cetuximab-MMAE failed to bind to B16 cells (Fig. 2f). Correlating with ADC target binding, CDX3379-

MMAE improved mouse survival of B16 tumor bearing mice while cetuximab-MMAE failed to improve mouse survival (Fig. 2e, right panel).

Summary Comment 3: “Also, the benefit of targeting with HER3 or ACPP should be demonstrated by comparison with free MMAE in comparable steady state levels.”

Response: While we agree in principle to compare free MMAE to targeted MMAE, dose limiting toxicities of free MMAE have precluded its use as a free drug in patients. Clinical development of MMAE is exclusively based on targeted delivery approaches of MMAE to tumors.

Summary Comment 4: “With the main trimodal warhead approach, an important control is missing.

Radiotherapy is increasingly used in combination with immune checkpoint inhibitors, with radiotherapy inducing formation of neo-antigens that evoke a T-cell infiltrate which can be activated by immune checkpoint inhibitors, as authors also refer to in the discussion. Without this important dual combination, a conclusion on the added value of targeted MMAE cannot be made.”

Response: As the Reviewer noted in our initial submission, we tested the dual combination of radiotherapy and anti-PD-1 only in the B16 cancer model but did not test this dual combination in the other models in Fig. 5. We have now included new data testing dual combination of anti-PD-1 immune checkpoint inhibition with IR or with MMAE to include all treatment permutations (Figs. 5b, 5c & 5d).

Specific Comment 1: “In fig 1b, the macrophage marker CD68 goes up in B16 after 5 nM MMAE for 72 hr. When looking at the IC50 (1 nM or so?), I suppose that after 72 hr there will be significant cell kill. What was the quality of RNA? Then, CD68 is involved in lysosomal trafficking and autophagy, a pathway that is however negatively enriched in the pathways. Authors should discuss this discrepancy. Another issue is that authors state that after MMAE treatment the angiogenesis pathway is enriched. Authors should rephrase this: angiogenesis involves endothelial cells and perivascular cells, therefore such statements can only be done in a tumor context, not in isolated cell lines.”

Response: We have corrected phrasing in the **Results** to state that MMAE treatment resulted in upregulation of genes involved in the angiogenesis process. As to the issue of RNA quality, NanoString analysis was run by the Core facility at UCSD where RNA quality was independently evaluated and passed QA testing prior to use in NanoString panels (Figure to right shows NanoString RNA quality from untreated control & MMAE samples). As to the point raised of CD68 and autophagy discrepancy, functional annotations of pathways was used as defined by NanoString for the IO 360 panel (found under Support Documentation on their website nanosttring.com). Genes grouped in their “Autophagy” pathway include: AKT1, BAD, BLC2, BCL2L1, BNIP3, DEPTOR, HIF1A, HMGB1, HRAS, KRAS, MAP3K7, MAPK10, MTOR, NRAS, PIK3CA, PIK3CD, PIK3R1, PIK3R2, PRKAA2, PRKACB, PTEN, RPS6KB1, RPTOR. CD68 is not included in their functional annotation for autophagy and can explain the discrepancy raised by the Reviewer.

Specific Comment 2: “CAL27 xenografts in immune compromised mice respond to anti-HER3 alone (Fig 2c) suggesting that the antibody also blocks intracellular HER3 signaling, independent from MMAE. Including MMAE now results in complete shutdown of growth of CAL27 tumors, but not of the CAL27 HER3 mutant ones. How can MMAE-resistance of a subpopulation of the mutant cells be explained? MMAE effects are completely downstream of the HER3 pathway. How do CAL27wt and mutant cells respond to targeted MMAE in vitro?”

Response: CDX3379 was designed as a HER3 blocking antibody and has anti-tumor effects. We used the CAL27 PIK3CA H1047R mutant cell lines to validate that our synthesized CDX3379-MMAE could bypass HER3 receptor signaling and deliver active MMAE to tumor cells resulting in tumor control. The Reviewer is correct that CAL27 PIK3CA H1047R mutant tumors did not respond as well as CAL27 WT tumors. We believe this is multi-factorial. First, CAL27 PIK3CA H1047R mutant cells grow faster and form tumor quicker in mice. Second, CAL27 PIK3CA H1047R cells are relatively resistant to radiotherapy in clonogenic cell assays

suggesting they may have improved DNA damage repair capabilities to cytotoxins. However in cell culture studies, we did not see an appreciable difference in MMAE sensitivity between the isogenic cell lines.

Specific Comment 3: “Line 266: fig 3e: the **** significance; is this IR+MMAE vs IR, or vs MMAE? Please clarify”

Response: Reviewer #2 raised a similar comment, (Rev #2, Comment 5). As mentioned, the statistical analysis in our original submission compared all 4 experimental groups. Based on Reviewers #2 & #4 suggestions, we have now updated statistical analysis in Fig. 3e. Importantly for scientific rigor & reproducibility, we would highlight that MMAE ± IR tumor response studies in syngeneic mice were replicated multiple times in different tumor models and using different drug carriers, i.e. ADC or ACPP conjugate (Figs. 2, 3, 4 & 5) which all consistently demonstrate the superiority of MMAE radiosensitization.

Specific Comment 4: “Experiments with RAG mice showing that CD8+ T-cells are involved in the response to MMAE+IR are intriguing. A mechanistic explanation is however lacking. If T-cells are crucial for tumor control under IR, why is there such good response in CAL27 xenografts in nude mice with monotherapy only?”

Response: We agree, the studies with RAG1 knockout mice were exciting in that they demonstrated a role for adaptive immunity on durable tumor control of syngeneic tumors treated with MMAE + IR. Using depleting antibodies, we then showed CD8 T cells are the necessary component of adaptive immunity mediated durable tumor control of MMAE radiosensitization. Taken together and with the published literature, we believe our studies highlight three mechanisms of action of MMAE tumor kill: 1) direct anti-tubulin cytotoxicity (Figs. 1a, 2c, 3a, 3b & 3c) 2) radiosensitization and clonogenic cell death (Figs. 3d, 3e, 4f & 4g) and 3) activating anti-tumor immune responses (Figs. 4 & 5). As discussed in our response to **Summary**

Comment 1, tumor xenografts are routinely eradicated by ADCs in pre-clinical models and our data is in line with the wealth findings in tumor xenografts. However, the clinical experience with ADCs has not been as robust. Moreover, a recent publication has also highlighted the modest anti-tumor efficacy of MMAE in syngeneic cancer models⁷. Our results provide a rational for combining MMAE armed ADCs with both radiotherapy and immunotherapy to maximize tumor kill through multiple mechanisms.

Specific Comment 5: “Line 122: Fig 1e relates to cell lines, not to xenografts as stated in the text”

Response: This has been clarified in the text that HER3 expression was assayed in cancer cells.

Specific Comment 6: “Line 183: please change title as here also immune-compromised models are tested.”

Response: Thank you, the title to section 2 of **Results** has been changed from “Anti-tumor efficacy of MMAE in immune-competent models” to “Efficacy of MMAE in murine cancer models”.

Specific Comment 7: “In syngeneic MOC tumors there is no activity of CDX3379-MMAE at all. How do MOC cell lines in vitro respond to CDX3379 and CDX3379-MMAE? Is HER3 in MOC cells a driver, or are there alternative kinases that converge in the same signaling pathways to rescue the CDX3379 effects.”

Response: In cell culture testing, MOC1 and MOC2 cells were sensitive to MMAE and CDX3379-MMAE as measured by Alamar Blue assay. For Alamar Blue assay cytotoxicity, we measured cell viability at 72 hr of drug exposure. We were limited in quantity of CDX3379 antibody and at nM dosing, CDX3379 exhibited minimal toxicity. Given that the scope of these studies focused MMAE radiosensitization in the tumor immune microenvironment, we limited cell culture studies and focused on the more unique aspects of our studies, namely MMAE radiosensitization to engage anti-tumor immune responses.

Reviewer #5

Summary Comment 1: “Key results and significance- The authors of the manuscript provide a detailed and comprehensive study that not only reports the synergistic effect of the MMAE-conjugates with IR for treating HER3 +ve tumours but have also detailed the mechanism that is necessary for immune-mediated responses, namely the role of the adaptive arm of immunity (through the need for functional CD8+ T cells) and the presence of checkpoint inhibition. The combination approach tested for tumour control was good evidence as a proof-of-concept study that advances the understanding of the principle behind how the tri-modal therapeutic strategy that was assessed works in vivo.

Data and Methodology- There were no notable concerns regarding the methodology that was followed and how they were described in the paper. The panel of results presented in the manuscript and the supplementary information was adequate. The interpretation of the results was also satisfactory and consistent. The data presented was thus convincing.”

Response: We truly appreciate these supportive comments of our work.

Summary Comment 2: “Analytical approach- The approach employed by the authors for the study was conceptually sound. The statistical tests employed were adequate and convincing.” Suggested improvements- The authors address the choice of antibody very well (lines 118-121) however it would also be relevant to briefly provide the reason behind the choice of the cell surface target HER3 over other well-characterized targets against which targeted antibodies are available as well, such as epidermal growth factor receptor (EGFR).”

Response: For proof-of-concept, we chose HER3 as our cell surface ADC target over other cell surface receptors for 2 reasons: 1) HER3 is a bona fide ADC target with HER3 targeted ADCs in clinical testing with cytotoxic warheads. The other clinically validated ErbB receptor targeted by ADCs is HER2. Given the expression of EGFR on normal cells, EGFR has not been pursued as an ADC target. 2) Celldex Therapeutics collaborated with our research group and generously provided clinical grade anti-HER3 antibody CDX3379. Clinical grade CDX3379 recognizes both human and murine isoforms of HER3 allowing us to simultaneously test our CDX3379-MMAE ADC in human xenografts in immune-deficient mice and syngeneic murine tumors grown in immune-competent mice. Given that CDX3379 is clinical grade can facilitate clinical translation of our work. We have included rationale in this revised manuscript.

Summary Comment 3: “Clarity and context- The manuscript was clear, concise, and adequately reports findings with respect to work reported in the past with one minor exception: 132-133, “There are limited pre-clinical studies evaluating the efficacy of MMAE within the immune context”; providing the specifics of what is known about MMAE-mediated immune modulation and the exact knowledge gap that the authors aim to address through the study would improve the clarity of the section.”

Response: Thank you for this suggestion. To our knowledge, there are no studies testing MMAE radiosensitization within the immune context or in combination with immunotherapies and so this was the primary knowledge gap that served as our motivation for the current set of studies. For manuscript flow, we have further expanded on this as mentioned by the Reviewer in the **Discussion**.

Summary Comment 4: “References- The authors of the manuscript have cited existing literature relevant to the study in a manner that was not only relevant but also comprehensive as well.”

Response: Thank you.

References

1. Hingorani, D.V. et al. Precision Chemoradiotherapy for HER2 Tumors Using Antibody Conjugates of an Auristatin Derivative with Reduced Cell Permeability. *Mol Cancer Ther* **19**, 157-167 (2020).
2. Wang, Z. et al. Disruption of the HER3-PI3K-mTOR oncogenic signaling axis and PD-1 blockade as a multimodal precision immunotherapy in head and neck cancer. *Nat Commun* **12**, 2383 (2021).
3. Buckel, L. et al. Tumor radiosensitization by monomethyl auristatin E: mechanism of action and targeted delivery. *Cancer Res* **75**, 1376-1387 (2015).
4. Adams, S.R. et al. Anti-tubulin drugs conjugated to anti-ErbB antibodies selectively radiosensitize. *Nat Commun* **7**, 13019 (2016).
5. Twyman-Saint Victor, C. et al. Radiation and dual checkpoint blockade activate non-redundant immune mechanisms in cancer. *Nature* **520**, 373-377 (2015).
6. Vanpouille-Box, C. et al. DNA exonuclease Trex1 regulates radiotherapy-induced tumour immunogenicity. *Nat Commun* **8**, 15618 (2017).
7. Boshuizen, J. et al. Cooperative targeting of immunotherapy-resistant melanoma and lung cancer by an AXL-targeting antibody-drug conjugate and immune checkpoint blockade. *Cancer Res* (2021).
8. Hingorani, D.V., Camargo, M.F., Quraishi, M.A., Adams, S.R. & Advani, S.J. Tumor Activated Cell Penetrating Peptides to Selectively Deliver Immune Modulatory Drugs. *Pharmaceutics* **13** (2021).

REVIEWERS' COMMENTS

Reviewer #1 (Remarks to the Author):

Dear Editor,

the authors provided new data and answered in a precise and exhaustive way to all the criticisms raised by the referees. The work has been greatly improved and therefore for this reviewer it deserves to be published in its current form.

Reviewer #2 (Remarks to the Author):

authors have satisfied my concerns and are congratulated to carry on this interesting study.

Reviewer #3 (Remarks to the Author):

The authors have performed additional experiments and satisfactorily addressed all my previous comments. I consider this body of work of the highest priority to the readers and valuable addition to the literature. A minor comment on the authors' response to my previous comment #7 - did the authors perform a one-time experiment with each of the B16, LL/2, and MC38 models or did the authors repeat three times for each of the aforementioned three implantable tumor models? If each of these tumor models was only done once, they probably need to be repeated at least once because implantable tumor models can vary considerably, recognizing that multiple models also support the rigor of the study.

Reviewer #4 (Remarks to the Author):

Authors show that targeted cytotoxic treatment with MMAE-conjugates of an HER3-antibody or an MMP-activatable cell penetrating peptide, combined with radiotherapy, induces tumor control in vivo via direct cell death combined with influx of T-cells. Authors show that this dual treatment is accompanied by upregulation of the immune checkpoint proteins. Combining MMAE-targeting with immune checkpoint inhibition results in good tumor control. The manuscript is of high interest because the individual modalities are already available or are in development for clinical use, allowing translation to clinical studies.

I congratulate the authors with this revised version. Substantial efforts were put in performing additional experiments to obtain further proof for the concept that direct cytotoxic targeting of cancer cells, further amplified by immune-mediated cell kill, may prevent outgrowth of treatment resistant clones.

Response Letter to Reviewers

Dear Reviewers,

Once again, we would like to thank you all for the time taken to review our revised manuscript that attempted to address the constructive criticisms raised by our initial submission. As you have noted, we worked diligently to address the initial critiques in a thorough fashion for this revision. We are delighted that your concerns have been adequately addressed and for your support of our work.

Reviewer #1

Comment: “the authors provided new data and answered in a precise and exhaustive way to all the criticisms raised by the referees. The work has been greatly improved and therefore for this reviewer it deserves to be published in its current form”

Response: We took to heart the initial comments by the Reviewers to help solidify our findings. We appreciate your support of our revised manuscript.

Reviewer #2

Comment: “authors have satisfied my concerns and are congratulated to carry on this interesting study.”

Response: I thank you for your constructive critiques during the review process that helped strengthen our studies.

Reviewer #3

Comment: “The authors have performed additional experiments and satisfactorily addressed all my previous comments. I consider this body of work of the highest priority to the readers and valuable addition to the literature. A minor comment on the authors' response to my previous comment #7 - did the authors perform a one-time experiment with each of the B16, LL/2, and MC38 models or did the authors repeat three times for each of the aforementioned three implantable tumor models? If each of these tumor models was only done once, they probably need to be repeated at least once because implantable tumor models can vary considerably, recognizing that multiple models also support the rigor of the study.”

Response: We thank you for your supportive comments and value you see in these studies. As to the issue you raise of reproducibility of the B16, LL/2 and MC38 models, we completely agree in the need to replicate findings of MMAE radiosensitization in syngeneic immune competent models as this is the 1st such report. The 3 models (B16, LL/2 and MC38) have been tested at least 3 times in the course of our experiments on MMAE radiosensitization in immune-competent models. As we continually refined dosing of MMAE and ionizing radiation, we have consistently found that the combination of MMAE + IR results in superior tumor control. Of note, initial pilot studies of ACP-MAE + IR were done in each of the cancer cell line models with a dose of 20 nmoles of ACP-MAE as opposed to our current optimized dose of 10 nmoles with similar results. Studies in the LL/2 model have been done in triplicate in comparing untreated, IR alone, ACP-MAE alone and ACP-MAE + IR. As noted, an initial study used ACP-MAE at 20 nmoles instead of our optimized dose of 10 nmoles. In the 2nd study, we used 3 Gy fractions of IR (this data was in our initial submission Fig. 3e) and in the 3rd experiment, we used 5 Gy per fraction (current Fig. 3e). The B16 model has been replicated >3 times as we have used the B16 model as our proving ground. The B16 model has also shown reproducible results of MMAE radiosensitized tumor control with MMAE conjugated to 2 different drug delivery platforms, i.e. ACP and ADC (Fig. 3e and Fig. 3h) and in studies with PD-1 (Fig 5b, e, f). The findings in MC38 model have also been replicated at least 3 times as shown in various studies in the manuscript (Fig 3e, 4h, 5c).

Reviewer #4

Comment: “Authors show that targeted cytotoxic treatment with MMAE-conjugates of an HER3-antibody or an MMP-activatable cell penetrating peptide, combined with radiotherapy, induces tumor control in vivo via direct cell death combined with influx of T-cells. Authors show that this dual treatment is accompanied by upregulation of the immune checkpoint proteins. Combining MMAE-targeting with immune checkpoint inhibition results in good tumor control. The manuscript is of high interest because the individual modalities are already available or are in development for clinical use, allowing translation to clinical studies. I congratulate the authors with this revised version. Substantial efforts were put in performing additional experiments to obtain further proof for the concept that direct cytotoxic targeting of cancer cells, further amplified by immune-mediated cell kill, may prevent outgrowth of treatment resistant clones.”

Response: Thank you for the supportive comments of our revised manuscript.